# L-RNA aptamer-based CXCL12 inhibition combined with radiotherapy in newly-diagnosed glioblastoma: dose escalation of the phase I/II GLORIA trial

The chemokine CXCL12 promotes glioblastoma (GBM) recurrence after radiotherapy (RT) by facilitating vasculogenesis. Here we report outcomes of the dose-escalation part of GLORIA (NCT04121455), a phase I/II trial combining RT and the CXCL12-neutralizing aptamer olaptesed pegol (NOX-A12; 200/400/600 mg per week) in patients with incompletely resected, newly-diagnosed GBM lacking MGMT methylation. The primary endpoint was safety, secondary endpoints included maximum tolerable dose (MTD), recommended phase II dose (RP2D), NOX-A12 plasma levels, topography of recurrence, tumor vascularization, neurologic assessment in neuro-oncology (NANO), quality of life (QOL), median progression-free survival (PFS), 6-months PFS and overall survival (OS). Treatment was safe with no dose-limiting toxicities or treatment-related deaths. The MTD has not been reached and, thus, 600 mg per week of NOX-A12 was established as RP2D for the ongoing expansion part of the trial. With increasing NOX-A12 dose levels, a corresponding increase of NOX-A12 plasma levels was observed. Of ten patients enrolled, nine showed radiographic responses, four reached partial remission. All but one patient (90%) showed at best response reduced perfusion values in terms of relative cerebral blood volume (rCBV). The median PFS was 174 (range 58-260) days, 6-month PFS was 40.0% and the median OS 389 (144-562) days. In a post-hoc exploratory analysis of tumor tissue, higher frequency of CXCL12+ endothelial and glioma cells was significantly associated with longer PFS under NOX-A12. Our data imply safety of NOX-A12 and its efficacy signal warrants further investigation.

Glioblastoma (GBM) is the most common malignant primary brain tumor in adults and is associated with a dismal prognosis[1]. Standard-of-care (SOC) treatment consists of maximum-feasible resection, external-beam radiotherapy (RT), and adjuvant therapy with temozolomide (TMZ)[2,3]. Tumors exhibiting unmethylated $O^6$-methylguanine DNA methyltransferase (MGMT) promoters are inherently resistant to chemotherapy[4], resulting in a median progression-free survival (PFS) of 4-5 months and a median overall survival (OS) of 10-15 months[5-8]. Beside age and clinical performance, the extent of tumor resection is an independent prognostic factor[9-11]. It is estimated that patients with completely resected tumors have a 61% higher likelihood of surviving 1 year than those with incomplete resection[12].

✉ e-mail: Frank.Giordano@umm.de; michael.hoelzel@ukbonn.de

Residual tumor cells, along with their unique post-radiotherapeutic tumor microenvironment, efficiently restore the therapy-depleted vasculature via vasculogenesis[13]. In contrast to angiogenesis, which is characterized by VEGF-mediated local sprouting of pre-existing vessels[14,15], vasculogenesis occurs de novo and is mediated by progenitor bone marrow-derived cells (BMDC)[16–21]. Recruitment of such BMDC occurs towards a gradient of the CXCR4/CXCR7 ligand CXCL12 (also known as SDF-1), which is induced through hypoxia-driven activation of HIF1α[17,22–24]. Besides its crucial role in attracting provasculogenic BMDC, CXCL12 is also suspected to, under certain circumstances, repel or sequester T cells[25–27], promote invasion of GBM cells, and decrease apoptosis[28,29].

Preclinical studies have shown improved intracranial tumor control in orthotopic GBM models after irradiation and subsequent inhibition of CXCR4 with the bicyclam derivative plerixafor[30]. Clinical safety of plerixafor after RT was recently reported in a phase I/II trial with 20 GBM patients as defined by the now outdated 2016 CNS WHO classification[31]. This trial also included patients with fully resected GBMs and more favorable molecular subtypes (IDH-mutant, MGMT methylated). Median PFS and OS compared favorably to historical data, and recurrence predominantly occurred outside of the irradiated areas, in line with the notion that CXCR4[+]/CXCR7[+] cells play a key role in restoration of the local post-radiation vasculature leading to recurrence[31]. As treatment was non-continuous, but abrogated after just 28 days, the effects of CXCR4 blockade may conceptually not have been fully exploited. Furthermore, its impact on poor-outcome GBM remains unclear. Targeting CXCL12 with the pegylated L-RNA aptamer olaptesed pegol (NOX-A12) was highly effective in an autochthonous rat brain tumor model mimicking a highly treatment-refractory GBM[32]. In this model, RT plus NOX-A12 significantly reduced tumor burden and resulted in sustained complete regressions. Due to its non-natural enantiomeric configuration, NOX-A12 harbors very low immunogenic potential[33] while exhibiting a high affinity and specificity to its target[34]. To assess the clinical safety and efficacy of RT combined with NOX-A12, we conducted an open-label, multicentric phase I/II trial.

Here, we report on the results of the dose escalation part of this trial and on post-hoc analyses of tumor tissue biomarker-dependent outcomes.

## Results

### Trial design, enrollment, and patient characteristics

The GLORIA trial (NCT04121455) is a multicentric phase I/II trial conducted to assess the safety and efficacy of RT combined with continuous i.v. treatment with NOX-A12 in newly diagnosed, incompletely resected, or biopsied GBM (CNS WHO grade 4) lacking MGMT promoter methylation (Fig. 1a). The trial consists of a completed NOX-A12 dose escalation part reported here, and multiple expansion arms with additional treatment schemes that are ongoing and, thus, not reported here. In the dose-escalation part, NOX-A12 was administered in a modified 3 + 3 rule-based design with escalating dose levels (DLs) of 200, 400, and 600 mg NOX-A12 per week. The primary endpoint of the trial was safety. Secondary endpoints included maximum tolerable dose (MTD), recommended phase II dose (RP2D), NOX-A12 plasma levels, topography of recurrence, tumor vascularization, neurologic assessment in neuro-oncology (NANO), quality of life (QOL), median PFS, 6-months PFS and OS. In addition, tumor tissue obtained during surgery was analyzed as an exploratory post-hoc analysis by multiplexed immunofluorescence (mIF) imaging.

Between September 2019 and September 2021, three patients were enrolled at each DL. One patient of DL 3 dropped out early and was replaced to ensure safety data quality, hence a total of ten patients were treated with RT and NOX-A12 (Fig. 1b). The median age at diagnosis was 65 years (range 43–79 years). Eight of ten patients had undergone partial resection, two were not amenable to resection and received biopsy only. Seven patients received normofractionated and three hypofractionated RT (Table 1).

### Pharmacokinetics and safety

NOX-A12 plasma levels reached a stable steady-state after approximately one week in all patients and surpassed 1.5 μM, which was considered to be the minimum plasma level required for disrupting CXCL12-mediated migration while minimizing bone marrow cell mobilization[34]. With increasing NOX-A12 DLs, a corresponding increase of NOX-A12 plasma levels was observed, in excess of CXCL12 levels (Fig. 2a). The median treatment time with NOX-A12 was 23.2 (12.3-48.1) weeks. Treatment was discontinued due to the end of treatment (EOT) as per protocol in two patients, suspected progressive disease (PD) in seven patients, and patient decision in one case. No patient discontinued treatment due to adverse events (AEs). In shared decision-making, the last patient enrolled continued treatment as per protocol beyond regular EOT after 26 weeks and also PD until a clinical deterioration in week 48. Treatment with RT and NOX-A12 was safe and well tolerated. No dose-limiting toxicities (DLTs) and no treatment-related deaths were observed. Thus, the MTD has not been reached and 600 mg per week of NOX-A12 is the RP2D also being taken forward into the ongoing expansion part of the trial. Out of 171 AEs, 13 (7.6%) were considered solely related to NOX-A12 (Table 2). Of all grade ≥2 adverse events (n = 84), 4 (4.7%) were considered to be NOX-A12-related, including one grade 3 AE at DL 3 (elevation of gamma-glutamyltransferase). Of note, this patient had idiopathic grade 1 elevated serum levels at baseline and suffered from an unrelated acute-on-chronic sigmoid diverticulitis soon after the AE. The majority of AEs were of grade 1 (50.9%) and were either unrelated or related to the GBM and RT. The most common treatment-emergent AEs (TEAEs, n = 160) were headaches which had been reported for a total of six patients with a maximum grading of grade 2 (Supplementary Fig. 1a). Increase of the alanine aminotransferase was the only treatment-related AE (TRAE) that occurred in three patients and did not exceed grade 2 (Supplementary Fig. 1b). Complete AE listings are summarized in Supplementary Data 1. TEAEs and TRAEs are provided in full in Supplementary Data 2–3 and Supplementary Tables 1–2, respectively.

### Radiographic response

All ten patients enrolled in the dose-escalation part of the trial were considered for the response analysis. As an exemplary responsive patient, C1-003 was treated with NOX-A12 continuously for 26 weeks as per protocol, reaching partial remission (PR) in week 9 and relapsed at the EOT as confirmed in a significantly aggravated MRI scan in week 33 (Fig. 2b). Under NOX-A12, nine patients (90%) showed radiographic response in terms of MRI lesion sizes in at least one timepoint of follow-up. Eight of the nine patients (89%) with target lesions (TLs) at baseline MRI assessment showed a TL response during NOX-A12 therapy, with four (44%) reaching PR as per radiologic mRANO criteria, i.e., ≥50% reduction in sum of the products of the longest perpendicular diameters (SPD) (Fig. 2c). Of these, two patients each were treated at DL 1 and DL 3, respectively. All three patients of DL 1 and all four of DL 3 reached ≥50% size reduction of at least one non-target lesion (NTL). In three cases, two at DL 1 and one at DL 3, at least one NTL disappeared completely (Fig. 2d). Advanced MRI parameters, including perfusion and diffusion assessment, were performed to investigate anti-vasculogenic effects. Under NOX-A12, all but one patient (90%) showed at best response reduced perfusion values in terms of relative cerebral blood volume (rCBV) (Supplementary Fig. 2) and threshold-calculated high fractional tumor burden (FTB[high]) with a median best response of −19.7% (24.0 to −55.5%) and −38.0% (9.3 to −100%) (Fig. 2e) indicating efficacy of the CXCL12 inhibitor therapy. In line with this, apparent diffusion coefficient (ADC) values were improved in 9 patients (90%) with a median best response of 29.2%

**a**

Inclusion criteria *
• Newly-diagnosed supratentorial GBM (WHO CNS grade 4)
• MGMT promoter unmethylated
• Incomplete resection/biopsy only
• ECOG ≤ 2

RT
60 Gy (2 Gy x 30)
40.05 Gy (2.67 Gy x 15)

NOX-A12
Continuous i.v. infusion at three DLs (200, 400, 600 mg/week)

Follow-up

Week 1          Week 6                                    Week 26 **

Safety monitoring
Advanced MRI (incl. perfusion/diffusion)

Primary endpoint:           Safety (incidence of adverse events)
Secondary endpoint:         NOX-A12 plasma levels, imaging parameters, PFS, OS *
Exploratory endpoints:      Multiplexed immunofluorescence imaging (CODEX®) - assessment of tissue biomarkers

**b**

Enrollment

Screened (n=12)

Screening failure (n=2)
• Patient withdrew consent (n=1)
• Decision of investigator (n=1)

Treatment

Enrolled (n=10)
DL 1 (3 pts)
DL 2 (3 pts)
DL 3 (4 pts)

Premature treatment
discontinuation (n=1)
• Decision of patient (n=1)

Completed treatment
with RT + NOX-A12
for ≥26 weeks (n=5)
for <26 weeks (n=4)

Follow-up

Completion of/
deceased during follow-up
(n=10)

Analysis

Analyzed for safety (n=10)

Analyzed for efficacy (n=10)

Included for translational analysis
(exploratory endpoint)
(n=10)

**Fig. 1 | Study outline of the GLORIA trial. a** Graphical overview of the study. GLORIA is a multicentric phase I/II trial conducted to assess the safety and efficacy of RT combined with escalating DLs of continuous i.v. treatment with NOX-A12 in newly diagnosed, incompletely resected, or biopsied GBM (CNS WHO grade 4) lacking MGMT promoter methylation ($n$ = 10). *A complete and more detailed list of eligibility criteria and outcome measures is provided under ClinicalTrials.gov Identifier: NCT04121455. **End of treatment: 26 weeks as per protocol; treatment continuation beyond 26 weeks per investigator's choice if the patient has clear clinical benefit. **b** Flow chart of the study. CODEX® CO-Detection by indEXing, DL dose level, ECOG Eastern Cooperative Oncology Group performance score, GBM glioblastoma, MGMT $O^6$-methylguanine DNA methyltransferase, MRI magnetic resonance imaging, NOX-A12 olaptesed pegol, OS overall survival, PFS progression-free survival, RT radiotherapy.

**Table 1 | GLORIA cohort patient characteristics (n = 10)**

| Variable | n (%) | Median (range) |
|---|---|---|
| Gender | | |
| Male | 7 (70) | |
| Female | 3 (30) | |
| Age (years) | | 65 (43–79) |
| ECOG score | | |
| 0 | 7 (70) | |
| 1 | 3 (30) | |
| Baseline NANO score | | 0 (0–4) |
| Resection status | | |
| Incomplete Resection | 8 (80) | |
| Biopsy | 2 (20) | |
| Methylation status | | |
| Methylated | 0 (0) | |
| Unmethylated | 10 (100) | |
| Residual tumor volume (cc) | | 4.3 (2.5–34.1) |
| Weeks post surgery | | 4.6 (3.9–7.1) |
| Tumor localization | | |
| Frontal lobe | 4 (40) | |
| Temporal lobe | 5 (50) | |
| Parietal lobe | 3 (30) | |
| Occipital lobe | 2 (20) | |
| Radiotherapy | | |
| Normofractionated | 7 (70) | |
| Hypofractionated | 3 (30) | |

(−15.2 to 161.8%) (Fig. 2f). NANO, QOL, and topography of recurrence were secondary endpoints but are not reported here.

## Clinical endpoints

The median PFS of the entire GLORIA cohort was 174 days (range 58–260 days), 6-month PFS 40.0%, and the median OS 389 days (144–562 days; Supplementary Fig. 3). This high variability prompted us to initiate a post-hoc exploratory analysis to search for potential biomarkers that correlate with NOX-A12 treatment responses focusing on CXCL12, the target of NOX-A12.

## Biomarker-dependent survival analysis

Analyzing publicly available single-cell RNA sequencing (scRNAseq) data from human GBM[35] showed the highest level and frequency of *CXCL12* mRNA expression in endothelial cells, followed by pericytes, myeloid (macrophages, microglia), and glioma cells (Fig. 3a, Supplementary Fig. 4). Therefore, we decided to assess total and cell-type specific CXCL12 protein expression in pre-treatment tumor samples obtained from GLORIA patients (n = 10) in a posthoc translational analysis. As an external control, pre-treatment tumor samples from an independent cohort of GBM patients with comparable clinical and histological features treated with SOC (n = 22; patient characteristics in Supplementary Table 3) were equally analyzed (Fig. 3b). We selected a panel of six antibodies validated for formalin-fixed paraffin-embedded (FFPE) tissue sections to identify endothelial cells (E; CD31), pericytes (P; α-SMA), macrophages (Mφ)/microglia (M; CD68), glioma cells (G; GFAP), proliferating cells (Ki-67) and CXCL12⁺ cells alongside 4′,6-diamidino-2-phenylindole (DAPI) as nuclear stain (Fig. 3c). For mIF imaging we employed co-detection by indexing (CODEX®), a well-established technology for profiling the tumor microenvironment of different tumor types including GBM[36,37]. Following image raw data processing, DAPI signals were used for automated nuclear segmentation with a custom-trained deep learning neural network algorithm

implemented within the HALO® AI analysis software. Subsequent cell-type assignment was based on marker expression (e.g., CD31 for endothelial cells). Lastly, the CXCL12 expression status was determined in a cell-type-specific manner (Supplementary Fig. 5, Methods). Samples of both the GLORIA and the SOC cohort were stained, imaged, and analyzed under the same conditions and settings, with all tumor areas being verified independently by two neuropathologists. Example images of CXCL12⁺ cell populations and H&E staining of the analyzed tumor areas are shown in Fig. 3c. All side-by-side illustrations of the analyzed areas in mIF staining and corresponding H&E staining are provided in Supplementary Fig. 6.

In total, we analyzed more than six million single cells with an average of 189,000 cells per sample (Supplementary Data 4). Consistent with scRNAseq data by Abdelfattah et al.[34], mIF revealed that the frequency of CXCL12⁺ cells was highest in endothelial cells (E12), followed by pericytes (P12), Mφ/microglia (M12) and glioma cells (G12) (Fig. 3d). As CXCL12 promotes post-radiogenic vasculogenesis and recurrence in preclinical models[30], we next asked whether CXCL12 positivity might be predictive for NOX-A12 treatment responses.

While a PFS event was definable for nine of the GLORIA patients, one patient was censored for PFS as per the statistical analysis plan (for details, see patient narratives in Supplementary Note 1). We noted a significant positive correlation between the frequency of CXCL12⁺ cells (total cells) and PFS in the GLORIA cohort (Spearman's rank correlation, $r_s = 0.712$, $p = 0.039$). This correlation was absent in the SOC cohort ($r_s = -0.251$, $p = 0.259$, Fig. 3e). Analyzing CXCL12 positivity per individual cell types, we detected significant positive correlations for frequency of CXCL12⁺ endothelial cells (E12; $r_s = 0.695$, $p = 0.046$) and of CXCL12⁺ glioma cells (G12; $r_s = 0.712$, $p = 0.039$) with PFS of patients enrolled in the GLORIA trial, while not reaching significance for frequency of CXCL12⁺ Mφ/microglia (M12; $r_s = 0.458$, $p = 0.223$) and CXCL12⁺ pericytes (P12; $r_s = 0.559$, $p = 0.126$) (Fig. 3f). Importantly, we found no significant correlations between any of the cell-type specific frequencies of CXCL12⁺ cells and PFS of the SOC cohort, including E12 ($r_s = 0.015$, $p = 0.946$) and G12 ($r_s = -0.261$, $p = 0.240$).

Both endothelial cells (E12) and glioma cells (G12) showed a significant correlation with PFS. While endothelial cells showed the highest relative CXCL12 positivity, the total number of endothelial cells was roughly twelve times lower than that of glioma cells. Therefore, we reasoned that combining E12 and G12 with approximately equal weights could embrace independent biological mechanisms and, thus, improve the correlation with NOX-A12 treatment responses. Consequently, we calculated the mean of the median-centered values of E12 and G12, resulting in a combined EG12 score that can hence adopt negative and positive values (Supplementary Fig. 7 and Supplementary Table 4). Here, the combined EG12 score strongly correlated with PFS ($r_s = 0.865$; $p = 0.005$; Fig. 4a) of the GLORIA patients. Again, in the SOC cohort, we found no significant correlation between the EG12 score and PFS ($r_s = -0.133$; $p = 0.556$; Fig. 4b). There was no significant correlation between the EG12 score and OS in the GLORIA cohort and the SOC cohort, but a positive and negative trend, respectively (Supplementary Fig. 8). Next, we used the EG12 score to divide the patients of the GLORIA and the SOC cohort by an unbiased median classifier into EG12ʰⁱᵍʰ and EG12ˡᵒʷ subgroups. With a median PFS of 183 vs. 92 days, EG12ʰⁱᵍʰ patients in the GLORIA cohort had a significantly longer PFS than EG12ˡᵒʷ patients (HR 0.12 (95% confidence interval (CI) 0.01–0.81); log-rank test, $p = 0.031$; Fig. 4c). We also detected a trend for prolonged OS for E12ʰⁱᵍʰ over E12ˡᵒʷ GLORIA patients (median OS 481 vs. 338 days; HR 0.25 (95% CI 0.03–1.17); log-rank test, $p = 0.075$; Fig. 4d). In the SOC cohort, no significant difference was measured in PFS (median PFS 118 vs. 136 days; HR 1.25 (95% CI 0.51–3.12); log-rank test, $p = 0.628$; Fig. 4e) or OS (median OS 328 vs. 288 days; HR 1.30 (95% CI 0.53–3.25); log-rank test, $p = 0.568$; Fig. 4f) between E12ʰⁱᵍʰ patients and E12ˡᵒʷ patients.

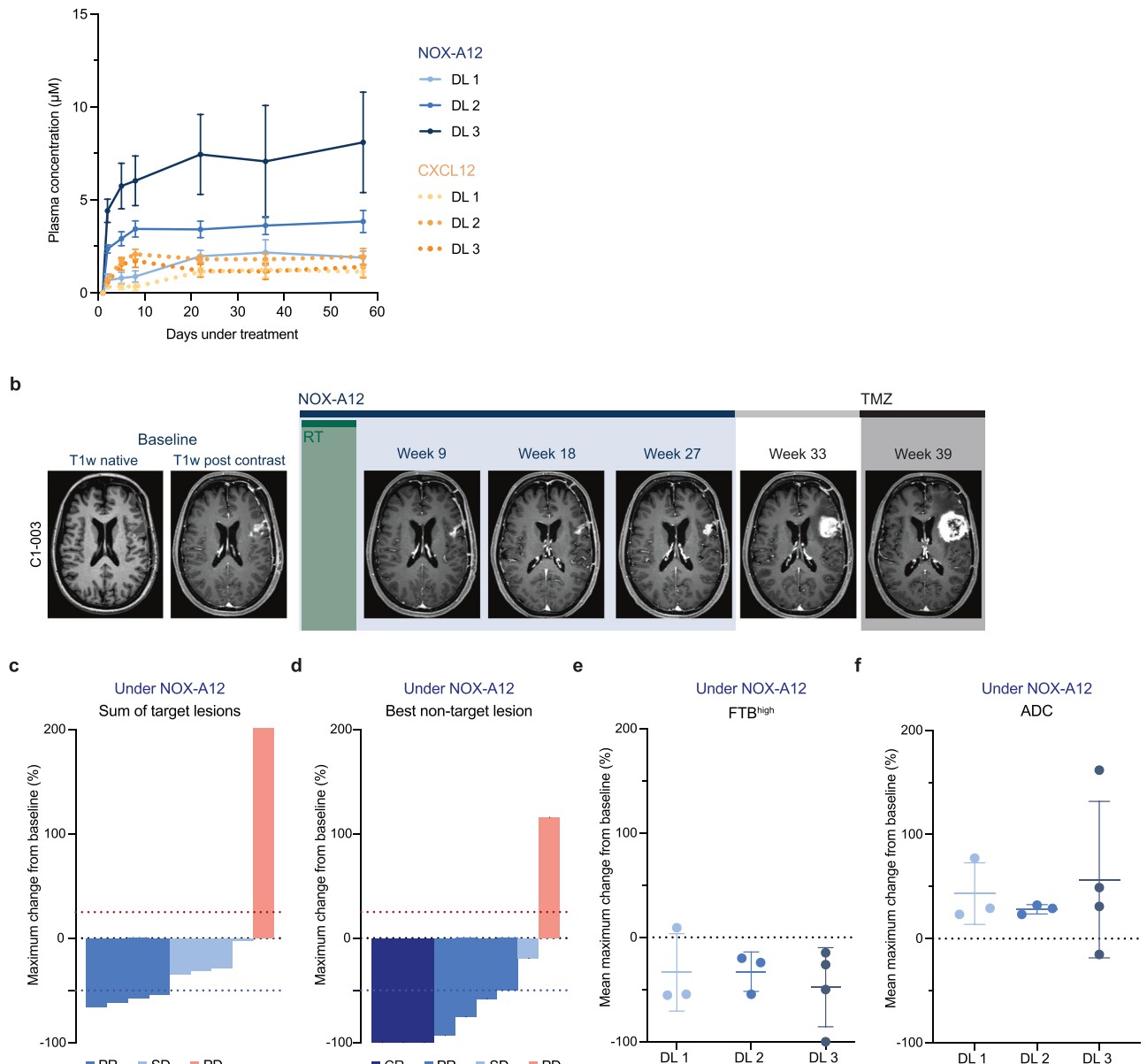

**Fig. 2 | Treatment with RT and NOX-A12 is safe and shows radiographic responses in conventional and advanced MRI. a** Serial plasma NOX-A12 (blue, full lines) and CXCL12 (orange, dashed lines) concentrations (μM) over treatment time (days) in respective GLORIA DLs ($n = 10$) indicated by color coding. Error bars indicate the standard error of the mean. **b** Representative illustration of the treatment course of a responding patient. Patient C1-003 was treated with RT (6 weeks; 2 Gy ad 60 Gy) and continuous NOX-A12 infusion for 26 weeks as per protocol, reaching partial remission in week 9. The patient relapsed at the end of NOX-A12 treatment (week 27) and deteriorated both before and after the initiation of TMZ. **c, d** Waterfall plots for best radiographic response as per mRANO under NOX-A12 (maximum change from baseline) of the sum of target lesion SPD (T1 Gd MRI) (**c**) and the best responding non-target lesion SPD (T1 Gd MRI) (**d**). Colors

from blue to red indicate CR, PR, SD, and PD for each patient. As per mRANO, red dotted line indicates 25% increase (PD), blue dotted line indicates −50% decrease (PR). **e, f** Dot plots depicting mean maximum change from baseline under NOX-A12 for FTB$^{high}$ (**e**) and ADC (**f**) of patients in the respective DLs (color-labeling in blue; 200 ($n = 3$), 400 ($n = 3$), 600 mg/week ($n = 4$)). Error bars indicate mean and standard deviation. Source data are provided as a Source Data file. ADC apparent diffusion coefficient, CR complete response, DL dose level, FTB$^{high}$ high fractional tumor burden, GBM glioblastoma, mRANO modified Criteria for Radiographic Response, NOX-A12 olaptesed pegol, PD progressive disease, PR partial response, RT radiotherapy, SD stable disease, SPD sum of product of perpendicular diameters, TMZ temozolomide.

## Discussion

In our study, we report the safety of RT and NOX-A12 in newly diagnosed, chemotherapy-resistant GBM meeting the primary endpoint of the trial. In addition, post-hoc tumor tissue analyses suggest improved clinical efficacy of this CXCL12-inhibiting L-RNA aptamer in a subgroup of patients characterized by a high frequency of CXCL12 positivity of endothelial and glioma cells.

Our trial supports previous findings that GBM recurrence after RT may be promoted by CXCL12-driven vasculogenesis[30,38–40]. We also demonstrated colocalization of CXCL12 with CD31+ endothelial cells and GFAP+ tumor (glioma) cells, in particular, identifying these cell populations as important sources of CXCL12. Our results are supported by a recent preclinical study that identified high endothelial CXCL12 expression as a key chemokine involved in pro-tumorigenic remodeling

**Table 2 | GLORIA trial adverse events by CTCAE grade and indication of relationships**

| Adverse events | n (%) |
|---|---|
| CTCAE grade | |
| Grade 1 | 87 (50.9) |
| Grade 2 | 59 (34.5) |
| Grade 3 | 24 (14.0) |
| Grade 4 | 1 (0.6) |
| Grade 5 | 0 (0) |
| Relationship | |
| No relationship | 81 (47.4) |
| Related to GBM | 43 (25.1) |
| Related to RT | 20 (11.7) |
| Related to RT & GBM | 6 (3.5) |
| Related to NOX-A12 | 13 (7.6) |
| yGT elevation | 1 (0.6) |
| ALT elevation | 3 (1.8) |
| Leukocytosis | 3 (1.8) |
| Constipation | 3 (1.8) |
| Dyspnea | 1 (0.6) |
| Paresthesia | 1 (0.6) |
| Pyrexia | 1 (0.6) |
| Related to NOX-A12 & GBM | 4 (2.3) |
| Related to NOX-A12 & RT | 2 (1.2) |
| Related to NOX-A12 & RT & GBM | 2 (1.2) |
| Total events | 171 (100) |

*ALT* alanine aminotransferase, *CTCAE* common terminology criteria for adverse events, *yGT* gamma-glutamyltransferase.

of the glioma microenvironment[41]. In addition, our approach also underscores the value of single-cell analyses at spatial resolution to identify microenvironmental mechanisms that determine disease prognosis and recurrence as recently shown[42,43].

Categorizing patients by their EG12 score in a low and high subgroup can potentially cause bias and overestimation of the observed effect[44]. Inherent with the design of early-phase clinical trials, only ten patients were treated with NOX-A12, and thus, our results will need further confirmation in larger cohorts. This future investigation will then also allow for gender-based assessments that were not feasible given the present patient numbers. The small cohort size may also explain why the difference between the OS of patients with EG12high versus EG12low tumors did not reach statistical significance. Specifically, since salvage treatments with variable efficiency[45,46] were not pre-specified in the trial protocol and hence individual management after recurrence varied from best supportive care only (three patients) to anticancer therapies, including TMZ (six patients), bevacizumab (five patients), CCNU (four patients), regorafenib (two patients), or re-irradiation (two patients), which may have impacted OS.

Insufficient crossing of the blood–brain barrier is a frequent limitation of novel drugs targeting brain tumors[47]. However, tissue penetration is not a prerequisite for NOX-A12 efficacy. NOX-A12 neutralizes CXCL12 in the blood, and it also releases and sequesters CXCL12 bound to glycosaminoglycans on the surface of tumor endothelial cells at the interface between the blood system and tumor cells[48]. Thereby, NOX-A12 disrupts CXCL12-dependent recruitment of circulating BMDC to the hypoxic tumor tissue, which prevents restoration of the tumor vasculature and hence restrains tumor cell growth in preclinical models[49,50]. Our study confirms in humans that NOX-A12 treatment indeed detaches and sequesters CXCL12, as we detected a profound accumulation of the

chemokine in the plasma reaching concentrations in the low micromolar range.

The mode of action of NOX-A12 strongly suggests that ongoing and uninterrupted treatment is crucial to prevent recurrence, as only the initial RT leads to devascularization of the tumor microenvironment, and an interruption of NOX-A12 infusions is likely to allow for rapid reconstitution of CXCL12 gradients and sequential vasculogenesis within few weeks[30]. Therefore, NOX-A12 effects on GBM control were possibly not fully exploited due to treatment interruptions or curtailments, especially in some of the responding patients. Notably, in some patients, NOX-A12 treatment was discontinued prematurely as a consequence of a misinterpretation of pseudo-progression (as observed and pathology-confirmed in patient C1-001).

The DLs selected for the GLORIA trial were supported by safety and efficacy considerations, as a NOX-A12 dose of 200 mg/week is expected to result in pharmacologically relevant mean plasma levels at steady state. Accordingly, NOX-A12 treatment resulted in excess of drug over target plasma levels in all DLs, which might explain the lack of a dose-dependency in this trial. The highest DL of 600 mg NOX-A12/week was safe and well tolerated. It is, therefore, the RP2D and also the DL being taken forward into expansion. While the GLORIA trial recruited only patients lacking MGMT methylation due to ethical reasons (no proven benefit of SOC with TMZ), there is no mechanistic reason to question the mode of action of NOX-A12 in MGMT methylated GBM. Thus, confirmation trials are now warranted that will continue to assess patient outcomes stratified by their EG12 score rather than other factors.

In conclusion, our results emphasize the need for further characterization of the GBM microenvironment to identify additional druggable targets and provide a rationale to intensify the in-depth investigation of a potential biomarker-stratified treatment of GBM with RT and CXCL12-directed therapy.

## Methods
### Trial design and oversight
GLORIA (SNOXA12C401, 2018-004064-62, NCT04121455) is a multicentric phase I/II study of RT in combination with NOX-A12 in first-line partially resected or unresected GBM (CNS WHO grade 4) patients with unmethylated MGMT promoter. The trial consisted of an initial dose-escalation arm (reported here) and additional expansion arms that evaluate NOX-A12 in combination with other drugs (follow-up ongoing). The first patient was enrolled on 23 September 2019. The last patient of the dose-escalation arm was enrolled on September 2, 2021.

The trial was first registered on EudraCT upon approval by the German authority (Bundesinstitut für Arzneimittel und Medizinprodukte (BfArM)) in May 2019 (https://www.clinicaltrialsregister.eu/ctr-search/trial/2018-004064-62/DE). The dose escalation was designed as a modified 3 + 3 rule-based design according to Le Tourneau et al.[51] with three successional cohorts consisting of three patients each. Patients of DL 1 were to be treated with a weekly dose of 200 mg, DL 2 with a weekly dose of 400 mg, and DL 3 with a weekly dose of 600 mg NOX-A12. After 4 weeks of treatment of the first patient of DL 1, the data safety monitoring board (DSMB) reviewed all DLTs, AEs, and relevant laboratory values. During the following ten weeks of treatment, the DSMB was kept informed continuously about all DLTs and SAEs, and, at the end of this period, reviewed all DLTs, AEs, and relevant laboratory values, including NOX-A12 plasma concentrations prior to enrollment of the next two patients of this DL. The evaluation was repeated prior to enrolling patients in DL 2 and after patients 2 and 3 received at least four weeks of treatment. The same procedures were performed prior to the enrollment of further patients in DL 2 and DL 3. If none of the three patients in any DL experienced a DLT, another three patients were to be treated at the next higher DL. However, if one of the three patients in a DL experienced a DLT, three more patients

 

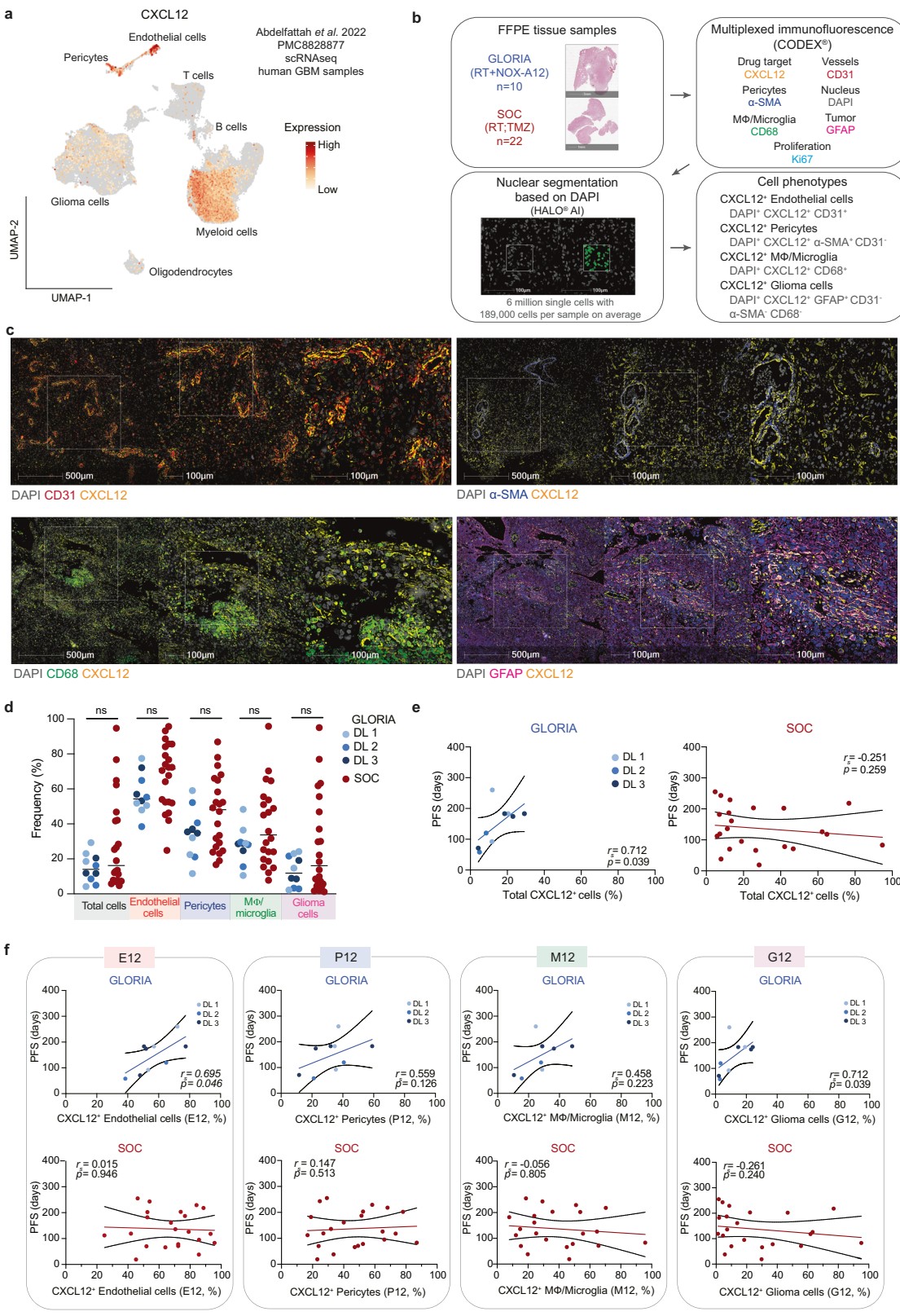

were to be treated at the same DL. The dose escalation was planned to be continued until at least two patients among a cohort of three to six patients experienced DLT (i.e., ≥33% of patients with a DLT at that DL), but the dose would not be escalated beyond 600 mg/week. The RP2D was defined as the DL just below this toxic DL, or 600 mg/week, if this DL is not toxic. DLTs, according to the common terminology criteria for adverse events (CTCAE, version 5.0), were defined as any grade 3–4

non-hematological toxicities (excluding grade 3 vomiting and/or nausea, if encountered without adequate and optimal prophylactic therapy), at any DL, assessed by the Investigator and/or the sponsor as related to NOX-A12.

Inclusion criteria of the dose-escalation arm of the trial were age ≥18 years, incompletely resected or biopsied GBM (detectable postoperative residual tumor), absence of MGMT promoter (hyper)

**Fig. 3 | CXCL12 positivity in endothelial cells and glioma cells correlates with PFS in the GLORIA cohort. a** UMAP projection overlaid with *CXCL12* mRNA expression in cell types from scRNAseq in human GBM samples (dataset from Abdelfattah et al.[35]). **b** Experimental setup of mIF imaging and outline of analysis pipeline. FFPE tissue samples were used for 7-plex mIF imaging; GLORIA cohort (RT + NOX-A12) (*n* = 10) and SOC cohort (RT; TMZ) (*n* = 22). All tumor areas were confirmed independently by two neuropathologists. Cell types and CXCL12 positivity were identified as indicated. **c** Representative images of GBM tissue samples from GLORIA cohort patients (*n* = 10) showing CXCL12 (yellow) expression in the cell types of interest: CD31+ endothelial cells (red); α-SMA+ pericytes (blue); CD68+ MΦ/microglia (green) and GFAP+ glioma cells (magenta). **d** Frequency of CXCL12+ cells per cell type measured in the GLORIA cohort (in blue; different DLs as indicated; *n* = 10) and in the SOC cohort (in red; *n* = 22). Unpaired two-tailed Mann–Whitney *U* test; ns: not significant (*p* > 0.05). **e** Spearman's rank correlation (*r*s) calculated between PFS (days) and total CXCL12+ cells (%) measured in the

GLORIA cohort (left; in blue and with DLs as indicated; *n* = 10) and in the SOC cohort (right; in red; *n* = 22). *r*s- and *p* values (two-tailed) are depicted in the corresponding graphs. **f** Spearman's rank correlation (*r*s) calculated between PFS (days) and CXCL12+ endothelial cells (%) out of total endothelial cells (E12), CXCL12+ pericytes (%) out of total pericytes (P12), CXCL12+ MΦ/Microglia (%) out of total MΦ/microglia (M12) and CXCL12+ glioma cells (%) out of total glioma cells (G12) measured in the GLORIA cohort (upper panels; in blue with DLs as indicated; *n* = 10) and in the SOC cohort (lower panels; in red; *n* = 22). *r*s- and *p* values (two-tailed) are depicted in the corresponding graphs. Source data are provided as a Source Data file. DL dose level, FFPE formalin-fixed paraffin-embedded, GBM glioblastoma, MΦ macrophages, mIF multiplexed immunofluorescence, NOX-A12 olaptesed pegol, PFS progression-free survival, RT radiotherapy, scRNAseq single-cell RNA sequencing, SOC standard-of-care, TMZ temozolomide, UMAP uniform manifold approximation, and projection.

methylation, Eastern Cooperative Oncology Group (ECOG) performance score ≤2, estimated life expectancy ≥3 months, stable or decreasing dose of corticosteroids and adequate hepatic and renal function. Sex was determined based on self-report. All patients were neuropathologically confirmed as GBM, IDH-wildtype (CNS WHO grade 4) according to the WHO classification for CNS tumors 2021 by immunohistochemistry (IHC). If patients were ≤54 years of age, they were assessed additionally by pyrosequencing for IDH1 and IDH2. GLORIA was conducted at six academic centers in Germany, whereas the protocol was approved by ethics committees at each participating site (ethic committees of the university hospitals of Mannheim, Bonn, Leipzig, Essen, Tübingen, and Münster). The study design and conduct complied with all relevant regulations regarding the use of human study participants. The trial followed the guidelines of the Declaration of Helsinki and the International Conference on Harmonization Good Clinical Practices Guidelines. Each patient provided written informed consent in accordance with established guidelines. The trial was reviewed by an independent data safety and monitoring committee. No trial participant received financial compensation. The sponsor agreed to the separate report of the dose escalation part of the trial as provided in this manuscript after the end of follow-up of the last patient of DL 3, as all patients in the expansion arm receive differing treatment combinations, limiting comparability. This dose escalation part was the only part of the trial in the initial protocol versions before the expansion arms were added to explore additional combination treatment options of interest. A minimally redacted version of the study protocol is provided in Supplementary Note 2.

### Treatment and endpoints
Following adequate cranial wound healing and implantation of a venous port catheter, treatment with NOX-A12 was initiated within six weeks post-cranial surgery. After an initial dose of 70, 160, or 230 mg per day, respectively, on day 1, patients were administered a fixed dose of 200, 400, or 600 mg NOX-A12 per week (DL 1, DL 2, DL 3) by continuous (24 h) i.v. infusion over a commercially available closed pump system (CADD®-Solis VIP Ambulatory Infusion Pump by Smiths Medical) starting on day 1. Treatment with NOX-A12 ended after 26 weeks. Patients with disease progression during the 26-week treatment period continued treatment with all assessments if deemed appropriate by the investigator. Continuation of treatment with NOX-A12 beyond 26 weeks was allowed as per each investigator's decision, if the patient had clear clinical benefit. No simultaneous systemic oncologic treatment was permitted. Baseline patient and treatment characteristics are enlisted in Supplementary Data 4. Clinical and radiographic follow-up assessments included standard and advanced magnetic resonance imaging (MRI) sequences. The primary endpoint of the trial was safety as per the incidence of AEs. Secondary endpoints included NOX-A12 plasma levels, MTD, RP2D, imaging parameters with a specific emphasis on monitoring re-

vascularization, topography of recurrence, PFS, OS, and clinician/patient-reported outcomes (CRO/PRO). Topography of recurrence as well as CRO and PRO (NANO, QOL) are not reported here, as analyses are planned after overall completion of the trial. As an additional exploratory endpoint, tumor tissue obtained in surgery was post-hoc analyzed by mIF staining (CODEX®). RT was initiated on day 2 after the start of NOX-A12 and administered as intensity-modulated, image-guided RT in a normofractionated (2 Gy per fraction) or hypofractionated (2.67 Gy per fraction) fashion up to cumulative doses of 60 or 40.05 Gy, respectively. For treatment planning, pre- and post-surgery MRI scans were co-registered on planning computer tomography (CT) scans. Gross tumor volumes (GTV), clinical target volumes (CTV), and planning target volumes (PTV) were defined as per current guidelines[52].

### Assessment of clinical and radiographic response
Patients visited the study site once weekly when presenting for the change of the medication cassette of the pump. Clinical routine follow-up visits included regular AE monitoring, physical and neurological assessments, vital signs, ECG, and blood tests. AEs were assessed and graded by the investigators according to the National Cancer Institute CTCAE, version 5.0. Baseline MRIs were obtained within a week prior to treatment initiation and up to 6 weeks post-surgery or post-biopsy. Follow-up MRIs were obtained every 8 weeks under treatment and at EOT. MRI imaging sequences included: 3D T1-weighted volumetric imaging (3D T1), T2-fluid-attenuated inversion recovery (FLAIR) imaging, diffusion-weighted imaging (DWI), T1-weighted dynamic contrast-enhanced perfusion imaging (DCE), T2-weighted turbo spin-echo imaging (T2 TSE), T2-weighted dynamic susceptibility contrast-enhanced perfusion imaging (DSC), and post-contrast 3D T1 imaging. The following additional advanced imaging parameters were calculated: DWI-derived ADC; diffusion susceptibility contrast (DSC)-derived leakage-corrected normalized rCBV and threshold-calculated FTB$^{high}$ (rCBV >1.75); DCE-derived transfer constant of contrast agent (Ktrans) between the blood and the extravascular extracellular space (EES), fractional EES volume ($v_e$), and fractional plasma volume ($v_p$). Following acquisition, MRI images were uploaded to a secure online portal (decidemedical, Clinflows) where a central quality check was performed. All image post-processing and interpretation were performed using IB Neuro™ (Imaging Biometrics), Olea Sphere (Olea Medical), and Mint Lesion™ (Mint Medical GmbH) software and assessed by a central reader not involved in the treatment of the patients (SB). MRI response values for all patients can be found in Supplementary Data 4.

### Outcome assessment
All MRI images were uploaded to an imaging database, and outcome was centrally assessed by a board-certified radiologist with expertize in the field blinded for study site and clinical status. Target lesions (TLs)

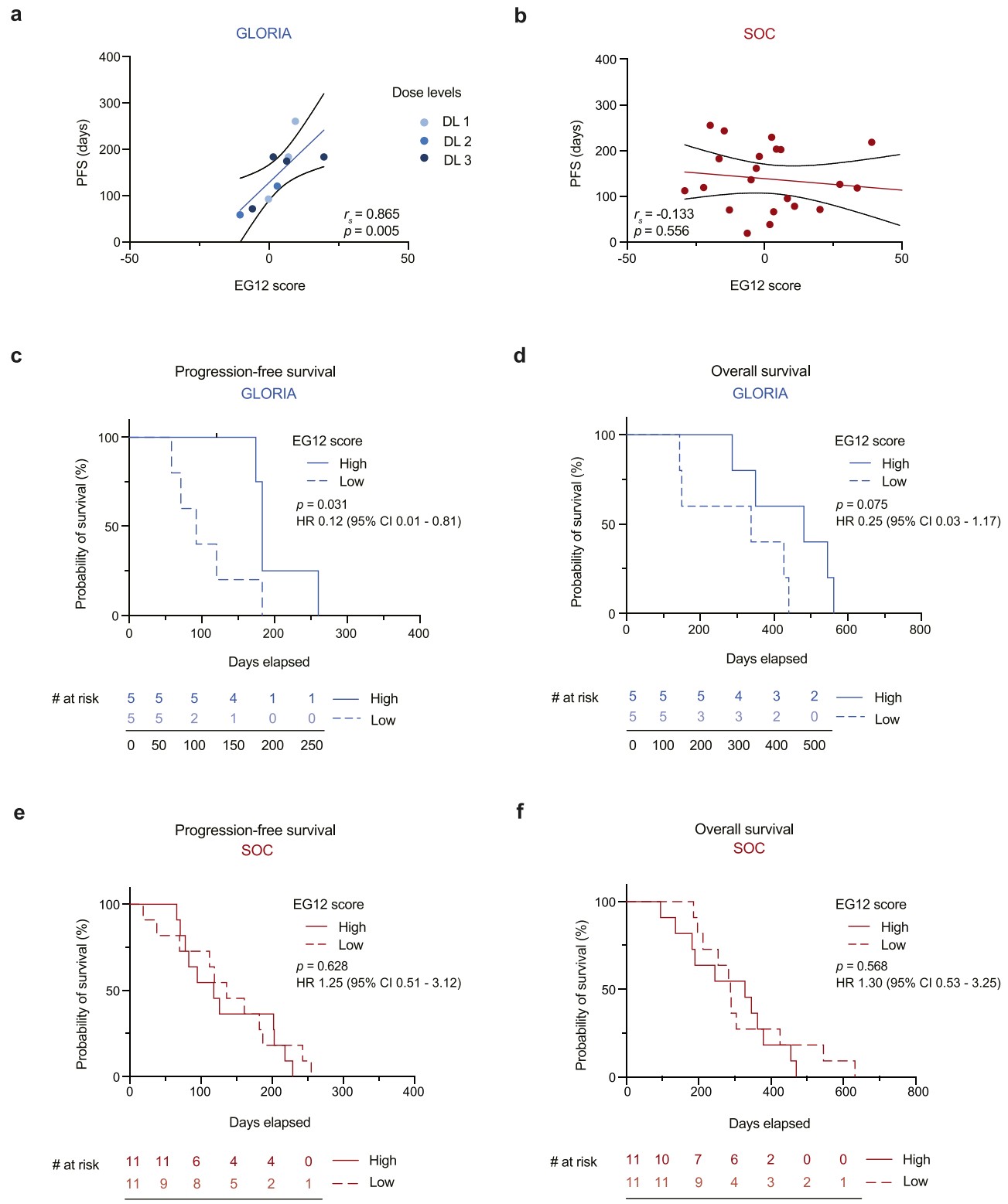

**Fig. 4 | EG12 correlates with PFS and is associated with improved survival in the GLORIA, but not in a SOC cohort. a**, **b** Correlation analysis of progression-free survival (days) with EG12 score in corresponding tumor tissue of the GLORIA cohort (*n* = 9; colors depict DL as indicated) (**a**) and the SOC cohort (*n* = 22) (**b**). Spearman's rank correlation ($r_s$); $r_s$- and *p* values (two-tailed) are depicted in the corresponding graphs. **c**, **d** Kaplan–Meier curves of progression-free (**c**) and overall survival (**d**) in days in the GLORIA cohort according to high (*n* = 5; continuous line) versus low (*n* = 5; dashed line) EG12 score. **e**, **f** Kaplan–Meier curves of progression-free (**e**) and overall survival (**f**) in days according to high (*n* = 11; continuous line) versus low (*n* = 11; dashed line) EG12 score in the SOC cohort. Log-rank test (two-tailed); *p* values are depicted in the corresponding graphs. Source data are provided as a Source Data file. CI confidence interval, DL dose level, HR hazard ratio, PFS progression-free survival, SOC standard-of-care.

and NTLs were identified, validated, and assessed in regard to tumor size (SPD) and corresponding timepoint tumor response according to the modified Criteria for Radiographic Response (mRANO)[53]. One patient enrolled had a singular residual tumor lesion meeting the inclusion criteria, while not qualifying for a TL (<10 mm in at least one diameter as per mRANO), thus documented as NTL. New non-measurable contrast-enhancing lesions only constituted progression in the case of complete response (CR). NTLs only impacted the response assessment in the case of a complete response of TLs. Preliminary tumor progression (PD) or regression (partial response, PR) required confirmation in a successive scan after 8 weeks. PFS was calculated as time (in days) between the first day of treatment with NOX-A12 to the day of PD. The PFS event was defined as the first date at which progression criteria had been met, i.e., (1) the date of the first sequentially confirmed MRI assessment resulting in preliminary PD or (2) the date of the radiographic assessment irrespective of its outcome in case of simultaneous investigator-assessed clinical progression attributable to no other cause apart from the tumor or; (3) the date of death by any cause if the patient died before clinical or radiographic progression. If preliminary PD was not confirmed in a sequential MRI and there was no subsequent SD, PR, or CR, the date of preliminary PD was still considered as an event for PFS if (1) the patient stopped protocol treatment due to clinical progression; (2) no further response assessments were done; or (3) the patient died due to any cause. For patients without a clinical or confirmed radiographic progression prior to a change of systemic therapy, PFS was censored at the date of initiation of a new anticancer treatment. Independent of MRI or clinical assessment, the diagnosis of PD was not established in the case of histopathologically confirmed pseudo-progression after re-surgery, which was the case in one patient where treatment with NOX-A12 was continued afterwards. OS was calculated as the time from the first day of treatment with NOX-A12 until death by any cause. The individual clinical courses, therapies, investigator decisions, and definitions of PFS and OS events for all patients are described in detailed narratives provided in Supplementary Note 1.

## SOC cohort

To benchmark tissue and outcome, we established a reference cohort of GBM patients treated outside of the study with SOC RT and optional TMZ at the University Hospital Bonn between 2010 and 2023. All patients had consented to analyses of preserved tissue and imaging studies. The procedures were approved by the Ethics Committee of the University Hospital Bonn (approval number: 222/23-EP). Histological criteria for selection of reference SOC patients were: newly-diagnosed GBM, IDH-wildtype (CNS WHO grade 4) according to the valid WHO classification for CNS tumors, absence of MGMT promoter methylation as confirmed by pyrosequencing[54]. Clinical criteria were: ECOG of 0–2, status post biopsy or incomplete resection, and first-line therapy with RT (and optionally TMZ, $n = 18/22$ receiving TMZ). In addition, despite leading to a possible (positive) survivorship bias, the availability of a baseline MRI scan and at least 2 consecutive scans suitable for mRANO assessment was mandatory for all SOC patients. The patient characteristics are provided in Supplementary Table 3. Sex was determined based on self-report. The PFS event was defined as the first date at which progression criteria had been met, i.e., the date of the sequentially confirmed MRI assessment resulting in preliminary PD as per mRANO, the date of initiation of second-line therapy or the date of death by any cause if the patient died before clinical or radiographic progression. PFS was defined as the time interval from the first day of RT to the PFS event. OS was calculated as the time from the first day of RT until death by any cause.

## Plasma NOX-A12 and CXCL12 concentrations

A liquid chromatography-UV assay, based on an anion-exchange chromatography analysis coupled to a UV detector, was used to detect and quantify the analyte NOX-A12 in patient plasma samples. The comparable calibration curve, generated from a standard-of-dilution series, corresponded to a linear range of NOX-A12 concentrations in human plasma from 0.5 to 200 µg/mL (0.034–13.6 µM). Quantification of CXCL12 concentrations in patient plasma samples was performed by a contractor (Swiss BioQuant AG) using HPLC-MS/MS bioanalytics. The comparable calibration curve corresponded to a linear range of CXCL12 (human SDF-1α and SDF-1β) concentrations in human plasma from 25 to 2500 nM.

## Buffers and solutions for multiplexed immunofluorescence

TCEP-reducing solution: 2.5 mM TCEP (Sigma, 646547) and 2.5 mM EDTA pH 8.0 (Invitrogen, AM9261) in ddH$_2$O, pH 7.0. Buffer C: 150 mM NaCl (Carl Roth, 9265.2), 2 mM Tris stock solution (Carl Roth, AE15.3), pH 7.2, 1 mM EDTA, and 0.02% w/v NaN$_3$ (AppliChem, A14300,1000) in ddH$_2$O. High-salt PBS: 900 mM NaCl in 1× DPBS (Gibco, 14190-094). CODEX® antibody stabilizer solution: 0.5 M NaCl, 5 mM EDTA, and 0.02% w/v NaN$_3$ in PBS antibody stabilizer solution (CANDOR Biosciences GmbH, 131125). Staining solution 1 (S1): 5 mM EDTA, 0.5% w/v bovine serum albumin (BSA, Carl Roth, 8076.3) and 0.02% w/v NaN$_3$ in 1× DPBS, stored at 4 °C. Staining solution 2 (S2): 61 mM NaH$_2$PO$_4$ (Sigma, S0876), 39 mM NaH$_2$PO$_4$ · H2O (Sigma, S9638), 250 mM NaCl in a 1:0.7 v/v solution of S1 and doubly-distilled H$_2$O (ddH$_2$O); final pH 6.8–7.0, stored at 4 °C. Staining solution 4 (S4): 0.5 M NaCl in S1, stored at 4 °C. Blocking buffer: S2 buffer containing B1 (1:20), B2 (1:20), B3 (1:20), and BC4 (1:15), stored at 4 °C. Blocking reagent 1 (B1): 1 mg/ml mouse IgG (Sigma, I5381) in S2, stored at 4 °C. Blocking reagent 2 (B2): 1 mg/ml rat IgG (Sigma, I4121) in S2, stored at 4 °C. Blocking reagent 3 (B3): sheared salmon sperm DNA (Invitrogen, AM9680), 10 mg/ml in H2O, stored at 4 °C. Blocking component 4 (BC4): Mixture of 57 non-modified oligonucleotides (Biomers) at a final concentration of 0.05 mM each in TE buffer (Sigma, 93302), stored at 4 °C (Supplementary Data 5). BS3 fixative solution: 200 mg/ml BS3 (ThermoFisher, 21580) in DMSO from a freshly opened ampoule (Sigma, D2650-5x5ML), stored at 20 °C in 3 µl aliquots. H2 buffer: 150 mM NaCl, 10 mM Tris pH 7.5, 10 mM MgCl$_2$ • 6 H$_2$O (Carl Roth, 2189.1), 0.1% w/v TritonTM X-100 (Sigma, X-100) and 0.02% w/v NaN$_3$ in ddH$_2$O. Plate buffer: H2 buffer containing DAPI nuclear stain (1:300, Biolegend, 422801) and 0.5 mg/ml sheared salmon sperm DNA. Fluorescent oligonucleotide stock solution (Biomers): 100 µM Fluorescent oligonucleotide dissolved in 1× TE buffer, stored in the dark at −20 °C. Fluorescent oligonucleotide working solution: Fluorescent oligonucleotide stock solution diluted 1:10 in 1× TE buffer, stored in the dark at 4 °C. Plate Buffer: H2 buffer containing DAPI nuclear stain (1:300) and 0.5 mg/ml sheared salmon sperm DNA.

## Multiplexed immunofluorescence of tumor tissue

All tumor samples were obtained following informed consent as part of SOC surgical procedures. All patients had consented to in-depth analyses of tissue. FFPE tumor samples were sliced by standard procedures at 3 µm slice thickness and adhered onto poly-L-lysine-coated coverslips. Antibody conjugation, tissue staining, and mIF imaging were performed (with modifications) as described elsewhere[36,55]. In short, purified, carrier-free antibodies were conjugated to maleimide-modified oligonucleotides (Biomers), concentrated, reduced, and washed with buffer. Maleimide-modified oligonucleotides were first dissolved in 1× DPBS, then added to the reduced antibody and incubated at room temperature for two hours in a 2:1 (w/w) ratio with the antibodies. Next, the conjugated antibodies were washed in high-salt PBS three times and then eluted by centrifugation at 3000 × $g$ for 2 min in the CODEX® antibody stabilizer solution. The conjugated antibodies were stored at 4 °C until usage. Prepared FFPE tissues were baked at 55 °C for 30 min, and rehydrated by immersion in fresh xylene, twice, for 5 min and in descending concentrations of ethanol, each step for 5 min (100% twice, 95% twice, 70%, ddH2O twice). Heat-induced

epitope retrieval was performed using 1× Dako target retrieval solution, pH 9 (Agilent) at high pressure, for 20 min. Tissues were then washed for 10 min in 1× TBS IHC wash buffer with Tween 20 (ThermoFisher, 28360). Tissues were blocked for 1 h at room temperature using 100 μl of blocking buffer. Conjugated antibodies were added to the blocking buffer, concentrated through a 50 kDa Amicon Ultra Filter, and resolved in the blocking buffer. Tissues were incubated with the antibody staining solution in a humidity chamber overnight at 4 °C. The following antibodies were used: Ki-67, 0.01 mg/ml, clone B56, BD Biosciences, Cat.# 556003 (RRID:AB_396287); SDF-1/CXCL12, 0.01 mg/ml, clone 79018, ThermoFisher, Cat.# MA5-23759 (RRID:AB_2608711); α-SMA, 0.01 mg/ml, clone 1A4, ThermoFisher, Cat.# 14-9760-82 (RRID:AB_2572996); CD31, 0.01 mg/ml, clone EP3095, Abcam, Cat.# ab226157; GFAP, 0.01 mg/ml, clone 2.2B10, ThermoFisher Scientific, Cat.# 13-0300 (RRID:AB_2532994); CD68, 0.005 mg/ml, clone KP-1, Biolegend, Cat.# 916104 (RRID:AB_2616797). The antibodies and their characteristics are additionally provided in a table overview in Supplementary Data 5 and the Reporting Summary. After staining, tissues were washed twice in S2 buffer and fixed with a three-step fixation process. First, tissues were fixed in S4 containing 1.6% paraformaldehyde (Electron Microscopy Science, 15710-S) for 10 min, followed by a 15 min-long incubation in 100% ice-cold methanol (Sigma, 34860-1L-R) for 5 min, and a final fixation with BS3 fixative solution dissolved in 1× PBS at room temperature for 20 min. Tissues were stored in S4 in a six-well plate at 4 °C for up to 2 weeks, or further processed for imaging. 400 nM fluorescent oligonucleotide working solution was aliquoted in Corning™ black 96-well plates (Merk, CLS3925-100EA) in 250 μl of plate buffer, according to the multi-cycle reaction panel. Image acquisition was performed on Zeiss Axio Observer 7 microscope equipped with a Colibri 7 LED Light source (Carl Zeiss), and a Prime BSI PCIe camera (Teledyne Photometrics). Imaging cycles were performed using an Akoya Phenocycler™ instrument and CODEX® instrument manager software (Akoya Biosciences). Automated images were acquired with the Plan-Apochromat 20×/0.8 M27 ($a = 0.55$ mm) objective (Carl Zeiss), and the imaging pipeline was controlled by a focus strategy with autofocus for each support point created, with a three z-stack image with a distance of 1.5 μm. DAPI (1:300 final concentration) was imaged in each cycle at an exposure time of 20 milliseconds and LED intensity of 40%. The images were processed with CODEX® Processor (Akoya Biosciences) and analyzed with HALO® Image Analysis software (Indica Labs, v.3.3). After each multi-cycle reaction, standard H&E staining was performed on the same tissue slice to confirm histopathological features. The H&E staining was analyzed independently by a neuropathologist with 5 years of experience and a board-certified neuropathologist with >30 years of experience in the field. With consensus, pathological features of GBM (CNS WHO grade 4) were again confirmed, and zonal characteristics within the tumor tissue were depicted. Adjacent regions like leptomeninges, hemorrhage, or healthy brain tissue were excluded, and only confirmed tumorous tissue parts were considered for the following analyses. Annotated H&E staining can be found in Suppl. Data 1. Analyses were performed using the Highplex FL module (v. 4.1.2) from HALO®. A nucleus/cytoplasm membrane % completeness threshold for positivity was set as follows: DAPI, Nucleus % Completeness Threshold 15%; CXCL12, Nucleus and Cytoplasm % Completeness Threshold 35%; CD68, Nucleus and Cytoplasm % Completeness Threshold 20%; CD31, Nucleus and Cytoplasm % Completeness Threshold 30%; Ki-67, Nucleus % Completeness Threshold 30%, α-SMA, Nucleus and Cytoplasm % Completeness Threshold 35%; GFAP, Nucleus and Cytoplasm % Completeness Threshold 30%. Cellular phenotypes were defined as follows: endothelial cells (DAPI⁺, CD31⁺), pericytes (DAPI⁺, α-SMA⁺, CD31⁻), Mφ/microglia (DAPI⁺, CD68⁺), tumor cells (DAPI⁺, GFAP⁺, CD68⁻, CD31⁻, α-SMA⁻). All phenotypes were also assessed for CXCL12 expression positivity. Nuclear segmentation was based on DAPI with a custom-trained deep learning neural network algorithm, with nuclear segmentation aggressiveness of 0.5 and nuclear size for positivity set between 12 and 1000 μm². Details on the multi-cycle reactions and oligo sequences can be found in Supplementary Data 5.

## scRNAseq analysis

Dataset GSE182109[35] was downloaded from the Broad Institute Single Cell Portal (https://singlecell.broadinstitute.org/single_cell). Data was analyzed and visualized using R version 4.2.2. *CXCL12* expression was overlayed on UMAP using scCustomize (Version 1.1.1)[56] using the FeaturePlot_scCustom() function with RColorBrewer (Version 1.1–3) and the color pallet "OrRd". Cells were called positive for *CXCL12* if they had a log-normalized count of 1 or more.

## Statistics and reproducibility

No formal sample size calculations were performed for this dose-escalation trial, and thus, no statistical method was used to predetermine the sample size. The dose escalation was designed as a modified 3 + 3 rule-based design as described above and in the study protocol provided in Supplementary Note 2. No data were excluded from the analyses. The experiments were not randomized, and thus, investigators were not blinded to allocation during experiments and outcome assessment. The independent central reader was blinded to all clinical aspects of the trial. Also, mIF was performed and analyzed blinded to all clinical aspects of the trial and patient identities. mIF sample sizes are provided for cohorts and subgroups.

Graphical elements were generated using GraphPad Prism 9 (GraphPad Software) and Adobe Illustrator 2023 (Adobe Inc.). Database management (eCRF) was carried out using Viedoc version 4.66 eCRF (Viedoc Technologies) and Microsoft Excel 2019 (Microsoft Corporation). Statistical tests were performed using GraphPad Prism 10 and R (V.3.3.2, x86_64-pc-linux-gnu) as specified in the figure legends. Descriptive statistics were applied to characterize the patient collectives, treatment response, and observed toxicity. Survival rates were estimated using the Kaplan–Meier method and statistically assessed by log-rank test and Cox proportional hazards regression. Group differences for continuous variables were evaluated using the unpaired two-tailed Mann–Whitney *U* test. Spearman's rank correlation was used for correlation analysis.

## Reporting summary

Further information on research design is available in the Nature Portfolio Reporting Summary linked to this article.

# Data availability

The study protocol is made available as Supplementary Note 2. The data generated in this study are provided in the Supplementary Information and Source Data file. High-resolution images of Supplementary Fig. 6 are provided in the following repository: "Giordano, Layer, Leonardelli et al. Supplementary Fig. 6", Mendeley Data, V1, https://doi.org/10.17632/wfhnv7j2wh.1 (https://data.mendeley.com/datasets/wfhnv7j2wh/1). The publicly available scRNAseq dataset used for re-analysis (from Abdelfattah et al.[35]) can be accessed via the GEO archive provided under accession ID GSE182109. Identifying individual participant data is protected and is not available due to data privacy laws. Individual de-identified participant data are available upon written request from the sponsor (according to local legal requirements for at least ten years). Source data are provided with this paper.

# Code availability

All code generated in this study to analyze and plot scRNAseq data has been deposited in the GitHub repository under accession code Giordano_Layer_Leonardelli_etal_CXCL12_GBM_GSE182109 (https://github.com/BaldLab).

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

## Acknowledgements

We wish to thank all patients and their families for their commitment to participate in this study. The clinical study was sponsored by TME Pharma AG (Berlin, Germany). TME Pharma AG also provided NOX-A12 clinical trial supply and partial funding for consumables related to this work. Parts of the translational research were supported by grants from the Deutsche Forschungsgemeinschaft (DFG, German Research Foundation; SFB 1389 UNITE Glioblastoma/TP-B05-404521405 to F.A.G.). J.P.L. was supported by a grant from *Novartis Stiftung für therapeutische Forschung* (foundation for therapeutic research) for personal equipment. S.L. was supported by post-doc research fellowships within the Mildred-Scheel School of Oncology Cologne-Bonn supported by the German Cancer Aid—Project ID 70113307 (Deutsche Krebshilfe) in which context expertize in mIF imaging was established for a project unrelated to this study. L.L.F. is supported by the BONFOR program of the Medical Faculty of the University of Bonn, Germany (grant ID 2022–1A-09). M.H. is a member of EXC2151 and supported by the Deutsche Forschungsgemeinschaft (DFG, German Research Foundation) under Germany's Excellence Strategy—EXC2151–390873048. We thank the study site staff for their ongoing support, particularly Christiane Landwehr, Katja Klever, Joana Kömpel, Mirco Muscheid, Monika Brüggemann, Inga Krause, Nadja Talhi, Sabrina Agkatsev, Gina Seidel. We thank Ute Heuser-Figgemeier and Alexandra Brüggemann from the Institute of Neuropathology and Sandra Bald and Simone Glees from the Department of Dermatology at the University Hospital Bonn for their help with the GBM tissue processing. The UKB histopathology research core facility is supported by the Deutsche Forschungsgemeinschaft (DFG, German Research Foundation) under Germany's Excellence Strategy—EXC2151–390873048. We would like to thank the Microscopy Core Facility of the Medical Faculty at the University of Bonn for providing instrumentation funded by the Deutsche Forschungsgemeinschaft (DFG, German Research Foundation)—project number 388168919.

## Author contributions

This study and its translational analysis were conceptualized and conceived by F.A.G., J.P.L., S.L., and M.H. F.A.G., J.P.L., T.Z., C.S., E.S., L.C.S., K.S., C.O., S.K., P.H., M.P., M.G., C.S., and U.H.-treated patients. S.B. and J.P.L. assessed and reviewed the patient's imaging. Material preparation and data collection were performed by J.P.L., S.L., and M.H. Histological staining was assessed by L.L.F. and T.P. mIF was established by S.L. and performed by S.L., J.P.L., and R.T. RNA sequencing data was analyzed by D.C., S.L., J.P.L., and M.H. Computational data analysis was performed by J.P.L. and S.L. F.A.G., U.H., and M.H. provided funding and resources. The first draft of the manuscript was written by J.P.L. and S.L. Tables and figures were created by J.P.L. and S.L. F.A.G., M.P., U.H., and M.H. reviewed and edited the data and manuscript. All authors commented on previous versions of the manuscript. All authors read and approved the final manuscript.

## Funding

## Competing interests

F.A.G. reports travel expenses, stocks and honoraria from TME Pharma AG related to this work; research grants and travel expenses from ELEKTA AB; grants, research grants, travel expenses and honoraria from Carl Zeiss Meditec AG; travel expenses and research grants from Varian Medical Systems, Inc.; travel expenses and/or honoraria from Bristol-Myers Squibb, Cureteq AG, Roche Pharma AG, MSD Sharp and Dohme GmbH, Siemens Healthineers AG, Varian Medical Systems, and AstraZeneca GmbH; non-financial support from Oncare GmbH and Opasca GmbH and patent US10857388B2 together with Carl Zeiss Meditec AG; all unrelated to this work. J.P.L. reports stocks and travel expenses from TME Pharma AG related to this work; travel expenses from Carl Zeiss Meditec AG, stocks and honoraria from Siemens Healthineers AG, and stocks from Bayer AG and BioNTech AG, all unrelated to this work. S.L. reports travel expenses from TME Pharma AG related to this work. C.S. has received speaker and/or advisory board honoraria from AbbVie, Bristol-Myers Squibb, HRA Pharma, Medac, Novocure, Roche, and Seagen not related to this work. E.S. reports travel expenses and honoraria for lectures from Carl Zeiss Meditec AG. C.O. reports travel support from Novocure; honoria by Horizon and Novocure and has received a Clinician Scientist Stipend of the University Medicine Essen Clinician Scientist Academy (UMEA) sponsored by the faculty of medicine and Deutsche Forschungsgemeinschaft (DFG). U.H. reports honoraria from Medac and Bayer AG, unrelated to this work. M.H. reports travel expenses, honoraria for webinars and research support (consumables) from TME Pharma AG related to this work. M.H. also reports honoraria from Bristol-Myers Squibb and Novartis unrelated to this work. A patent application related to biomarker identification has been filed by F.A.G., J.P.L., S.L., and M.H. (EP23000076.2; EP23000075.4). The remaining authors declare no competing interests.

## Additional information

**Frank A. Giordano**[1,2,13] ✉, **Julian P. Layer** [3,4,13], **Sonia Leonardelli**[4,13], **Lea L. Friker** [4,5], **Roberta Turiello** [4], **Dillon Corvino**[4], **Thomas Zeyen**[6], **Christina Schaub**[6], **Wolf Müller**[7], **Elena Sperk** [1], **Leonard Christopher Schmeel** [3], **Katharina Sahm**[2,8,9], **Christoph Oster** [10], **Sied Kebir**[10], **Peter Hambsch**[11], **Torsten Pietsch**[5], **Sotirios Bisdas** [12], **Michael Platten** [2,8,9], **Martin Glas**[10], **Clemens Seidel** [11], **Ulrich Herrlinger** [6,13] & **Michael Hölzel** [4,13] ✉

[1]Department of Radiation Oncology, University Medical Center Mannheim, Medical Faculty Mannheim, University of Heidelberg, Mannheim, Germany. [2]DKFZ-Hector Cancer Institute at the University Medical Center Mannheim, Mannheim, Germany. [3]Department of Radiation Oncology, University Hospital Bonn, University of Bonn, Bonn, Germany. [4]Institute of Experimental Oncology, Medical Faculty, University Hospital Bonn, University of Bonn, Bonn, Germany. [5]Institute of Neuropathology, University Hospital Bonn, University of Bonn, Bonn, Germany. [6]Department of Neurooncology, Center for Neurology, University Hospital Bonn, Bonn, Germany. [7]Institute of Neuropathology, University Hospital Leipzig, University of Leipzig, Leipzig, Germany. [8]Department of Neurology, Medical Faculty Mannheim, MCTN, Heidelberg University, Mannheim, Germany. [9]DKTK Clinical Cooperation Unit Neuroimmunology and Brain Tumor Immunology, German Cancer Research Center, Heidelberg, Germany. [10]Division of Clinical Neurooncology, Department of Neurology, Center for Translational Neuro- and Behavioral Sciences (C-TNBS) and West German Cancer Center, German Cancer Consortium, Partner Site Essen, University Hospital Essen, University Duisburg-Essen, Essen, Germany. [11]Department of Radiation Oncology, University Hospital Leipzig, University of Leipzig, Leipzig, Germany. [12]Lysholm Department of Neuroradiology, University College London, London, UK. [13]These authors contributed equally: Frank A. Giordano, Julian P. Layer, Sonia Leonardelli, Ulrich Herrlinger, Michael Hölzel. ✉e-mail: Frank.Giordano@umm.de; michael.hoelzel@ukbonn.de

