## [Peer Review File · Nature Communications]

Reviewers' Comments:

Reviewer #1:

Remarks to the Author:

The authors present a well-written manuscript on the safety dose/escalation cohort of CXCL12 inhibition combined with radiotherapy in newly diagnosed GBM.

1. Although it is assumed, are the patients all IDH-WT GBM as per the WHO 2021?
2. The escalation design is not named, and only 3 (planned) patients were examined per dose level. Can the authors explain why a more traditional type of dose escalation study was performed?
3. Although the score derived as a biomarker may be meaningful - the exceptionally small sample size and lack of evaluation/training diminish this reader's excitement.
4. Further, categorizing the score into high and low groups only adds bias to bias. Please see: PMID: 33579807
5. The legend for Fig 4 is misleading - the score is not shown to predict response - it is associated with survival in a small cohort and not in another. To show that the score predicts response, a model with an interaction term and both cohorts would be needed. As well as a larger sample size.

Reviewer #2:

Remarks to the Author:

In this manuscript, the authors, conducted a Phase 1, dose escalation study in 10 patients with newly diagnosed GBM. Patients received radiation and NOX-A12 which targets CXCL12, a molecule important in driving tumor vasculogenesis. The authors enrolled patients with GBM who have some of the worse prognostic characteristics (biopsy/subtotal resection and MGMT gene promoter unmethylated). The strengths of the manuscript include that it was a well conducted study with inclusion of a standard of care comparison cohort and inclusion of tissue correlative analyses that possibly identified a potential biomarker of ontarget response to the drug. NOX-A12 was not associated with significant toxicity. My biggest concern is that they report 9/10 patients had a radiographic response, but it is not clear to me this is true as outlined below. Overall, this is an incremental advance to the literature due to the small sample size and early stage of the trial.

Major Concerns

- Fig. 2c is not convincing for a PR. In the baseline scan, there is the surgical cavity without clear evidence of measurable disease around the surgical cavity. The follow up image from week 9 just looks like the surgical cavity collapsed with expected evolving post surgical changes. Based on the images shown, this would not constitute a PR since there was no measurable disease at baseline (surgical cavity should not be included in the tumor measurement).
- Two patients were <50 years old so how was screening for IDH mutation done - IHC or was next generation sequencing performed?

Minor Concerns

- The sentence "Given that endothelial cells showed the highest relative CXCL12 positivity at roughly twelve times lower total cell numbers compared to the glioma cells..." (line 179-180) is confusing and hard to understand.
- Was the SOC cohort a contemporaneous cohort? Please include the years during which the SOC cohort was identified.

Reviewer #3:

Remarks to the Author:

Overall summary

This is a well written report of the safety and outcome of the first 10 patients enrolled onto the Phase 1 dose escalation component of the GLORIA trial of a CXCL12-neutralizing aptamer given in combination with radiation. The authors note promising outcomes in this rather poor prognosis subgroup of glioblastoma and identify a subgroup responsive to this therapy.

Comments

1) Safety - 10 patients were enrolled onto 3 dose levels. It is unclear what dose escalation methodology was used to inform dose escalation (3+3 or Bayesian or other) but this will need clear description, particularly given safety was the primary outcome of the the Phase 1 component of the trial reported. Methodology and DLT criteria will need to be provided in a revision. The authors provide a list of treatment emergent adverse events in the supplemental data, however this table need clarification. I note 7 events of seizure, and 2 events of status epileptus in the data listing; alongside events of brain oedema, and psychomotor disturbance. The authors will need to clarify why these events were not considered 'possibly' related to novel treatment, given the temporal occurrence and how the authors concluded that NOX-A12 did not contribute to neurological AEs. It would be more helpful to provide tables of listings of TEAE and TRAE per dose level, and commenting specifically on AEs that are occurring >5%.

2) Pharmacokinetics - the authors do not comment on brain penetration of the aptamer, at least from preclinical data. Is there expected variability from the 3 dose levels tested that may impact efficacy?

3) Efficacy - the study is laudable in recruiting a poor prognosis group of patients with incompletely resected GBM, with unmethylated MGMT. It is noted that 10 patients is an extremely small number to be drawing more than exploratory conclusions, however given the poor prognosis of patients, it does appear to be promising that 8/9 has some response in target lesions. Line 129 - there is no established radiological criteria to define PR in non-target lesions so this sentence appears to be overcalling and should be removed.

4) Biomarker translational work - the sample size is small for conclusive subgroup analyses, however the study does convincingly demonstrate proof of concept for mechanism of action with patients with higher expression of CXCL12 benefitting more.

Overall, this manuscript warrants publication with minor corrections as described above. The authors will need to provide study protocol, along with dose finding schema and DLT criteria and expand on the safety and tolerability description as above. Which dose level is being taken forward into expansion?

The proof of concept and preliminary efficacy in a very poor prognosis subset of GBM is exciting and warrants further exploration.

We would like to thank the reviewers and the editors for taking the time to thoroughly review our manuscript and for providing valuable feedback. We believe that our manuscript substantially gained in quality following their suggestions. We are confident that our findings will further vitalize the research on CXCL12 inhibition and mark a first step towards further and more profound assessment of the combination of radiotherapy and CXCL12 inhibition as a treatment option for selected patients with glioblastoma.

We addressed all the comments and suggestions in our point-by-point response. All changes to the manuscript are color highlighted. With regard to central points raised by the reviewers and editor, we now provide detailed further methodological information and background about the trial design. In our revised manuscript, we also focused on the remarks regarding the radiographic response analyses. In addition, we tempered done the interpretation of the exploratory endpoint (EG12 score) recognizing the small cohort size inherent to our phase I/II clinical trial.

Reviewer #1

The authors present a well-written manuscript on the safety dose/escalation cohort of CXCL12 inhibition combined with radiotherapy in newly diagnosed GBM.

We thank the reviewer for the kind appreciation of our work.

1. Although it is assumed, are the patients all IDH-WT GBM as per the WHO 2021?

All patients were confirmed as glioblastoma, IDH-wildtype (CNS WHO grade 4) according to the WHO 2021 classification for CNS tumors by immunohistochemistry. If patients were younger than 54 years, they were routinely assessed additionally by pyrosequencing for IDH1 and IDH2. To clarify, we made the following modification to the Methods section of the manuscript:

Page 10, paragraph 3: *All patients were neuropathologically confirmed as glioblastoma, IDH-wildtype (CNS WHO grade 4) according to the WHO classification for CNS tumors 2021 by immunohistochemistry (IHC). If patients were ≤ 54 years of age, they were assessed additionally by pyrosequencing for IDH1 and IDH2.*

2. The escalation design is not named, and only 3 (planned) patients were examined per dose level. Can the authors explain why a more traditional type of dose escalation study was performed?

The design of the dose escalation is a traditional 3+3 rule-based design as described by Le Tourneau et al. This design has the advantage of being easy to implement and safe, which was our motivation to use this approach.

We added a short description to the Results section and now provide detailed information about the escalation design in the Methods section:

Page 5, paragraph 1 (Results section): *In the dose-escalation part of the trial reported here, NOX-A12 was administered in a 3+3 rule-based design with escalating dose levels (DLs) of 200, 400 and 600 mg NOX-A12 per week.*

Page 10, paragraph 2 (Methods section): *The dose escalation was designed as a 3+3 rule-based design according to Le Tourneau et al [Le Tourneau et al.] with three successional cohorts consisting of three patients each. Patients of DL 1 were to be treated with a weekly dose of 200 mg, of DL 2 with a weekly dose of 400 mg and of DL 3 with a weekly dose of 600 mg NOX-A12. After four weeks of treatment of the first patient of DL 1, the data safety monitoring board (DSMB) reviewed all DLTs, AEs, and relevant laboratory values. During the following ten weeks of treatment, the DSMB was kept informed continuously about all DLTs and SAEs, and, at the end of this period, reviewed all DLTs, AEs, and relevant laboratory values including NOX-A12 plasma concentrations prior to enrolment of the next two patients of this DL. The evaluation was repeated prior to enrolling patients in DL 2 and after patients 2 and 3 received at least four weeks of treatment. The same procedures were performed prior to enrolment of further patients in DL 2 and for DL3. If none of the three patients in any DL experienced a DLT, another three patients were to be treated at the next higher DL. However, if one of the three patients in a DL experienced a DLT, three more patients were to be treated at the same DL. The dose escalation was planned to be continued until at least two patients among a cohort of three to six patients experienced DLT (i.e., $\geq 33\%$ of patients with a DLT at that DL), but the dose would not be escalated beyond 600 mg/week. The recommended dose for phase II trials was defined as the DL just below this toxic dose level, or 600 mg/week if this DL is not toxic.*

3. Although the score derived as a biomarker may be meaningful - the exceptionally small sample size and lack of evaluation/training diminish this reader's excitement.

We are aware of the limitations of a phase I/II clinical trial and made several adjustments towards a more nuanced presentation of the biomarker and edited the manuscript in the Abstract, Results and Discussion sections as follows:

Page 3, paragraph 1 (Abstract): *Our data imply clinical efficacy of NOX-A12 in EG12^{high} GBM and suggest further evaluation of the EG12 score as a potential predictive biomarker in CXCL12-targeting therapies.*

Page 8, paragraph 2 (Discussion): *By showing that EG12 is associated with response to RT and CXCL12 inhibition, we also provide evidence on the mode of action of the combinatory treatment.*

Page 9, paragraph 4 (Discussion): *In conclusion, our results emphasize the need for further characterization of the GBM microenvironment to identify additional druggable targets and provide a rationale to intensify in-depth investigation of a potential biomarker-stratified treatment of GBM with RT and CXCL12-directed therapy.*

Page 22, paragraph 1 (Figure caption of Figure 4): *EG12 correlates with PFS and is associated with improved survival in the GLORIA, but not in a SOC cohort.*

4. Further, categorizing the score into high and low groups only adds bias to bias. Please see: PMID: 33579807

We understand the concerns of the reviewer and moderated the strength of our statements as described above (#3). We also included the concern about categorizing scores in the Discussion section of the revised manuscript:

Page 8, paragraph 4: *Categorizing patients by their EG12 score in a low and high subgroup can potentially provoke bias and cause overestimation of the observed effect [Polley et al.]. Inherent with the design of early phase clinical trials, only ten patients were treated with NOX-A12, and thus our results will need further confirmation in larger cohorts.*

Nevertheless, we think that the underlying biological rationale and the comprehensible proof of the drug's mode of action provide a clear rationale to assume an effect dependency on the defined patient subgroups. Therefore, we still see value in reporting our results on EG12 subgroups as a potential strategy how to stratify patients in future studies.

5. The legend for Fig 4 is misleading - the score is not shown to predict response - it is associated with survival in a small cohort and not in another. To show that the score predicts response, a model with an interaction term and both cohorts would be needed. As well as a larger sample size.

We adjusted the figure caption as suggested:

Page 22, Fig. 4: *EG12 correlates with PFS and is associated with improved survival in the GLORIA, but not in a SOC cohort.*

Reviewer #2

In this manuscript, the authors, conducted a Phase 1, dose escalation study in 10 patients with newly diagnosed GBM. Patients received radiation and NOX-A12 which targets CXCL12, a molecule important in driving tumor vasculogenesis. The authors enrolled patients with GBM who have some of the worse prognostic characteristics (biopsy/subtotal resection and MGMT gene promotor unmethylated). The strengths of the manuscript include that it was a well conducted study with inclusion of a standard of care comparison cohort and inclusion of tissue correlative analyses that possibly identified a potential biomarker of ontarget response to the drug. NOX-A12 was not associated with significant toxicity. My biggest concern is that they report 9/10 patients had a radiographic response, but it is not clear to me this is true as outlined below. Overall, this is an incremental advance to the literature due to the small sample size and early stage of the trial.

We thank the reviewer for acknowledging our work.

Major Concerns

- Fig. 2c is not convincing for a PR. In the baseline scan, there is the surgical cavity without clear evidence of measurable disease around the surgical cavity. The follow up image from week 9 just looks like the surgical cavity collapsed with expected evolving post surgical changes. Based on the images shown, this would not constitute a PR since there was no measurable disease at baseline (surgical cavity should not be included in the tumor measurement).

For clarification, we would like to point out that our baseline scans do not correspond to an immediate post-surgery MRI, but reflect the situation at treatment initiation (day -7 to day -1) which may differ (per protocol) up to 6 weeks from a post-surgery MRI.

Together with the central reader of our trial, we re-assessed the MRI images of the patient shown in Fig. 2c confirming PR of this patient: While this is a patient with only a smaller residual tumor of 3.3 cm³ at baseline, it was clearly measurable. We now replaced the previous axial images with new ones from a different slice that provides a better visibility of the initial residual tumor and a T1w native image for side-by-side comparison with T1w post contrast of the baseline scan (**revised Fig. 2c**).

Regarding methodology: After an initial quality check, a central reader from a contracted third-party provider site not involved in the treatment of study patients and a recognized expert in the field assessed all MRIs acquired and post-processed using a highly standardized routine (outlined in the imaging manual) using a set of classical analysis softwares. The RANO recommendations were strictly followed by the central reader.

To clarify our efforts to ensure imaging acquisition and readout quality for the readers, we extended our statements in the Methods section:

Page 11, paragraph 3: *Baseline MRIs were obtained within a week prior to treatment initiation and up to six weeks post-surgery or post-biopsy.*

Page 12, paragraph 1: *Following acquisition, MRI images were uploaded to a secure online portal (decidemedical, Clinflows) where a central quality check was performed. All image post-processing and interpretation was performed using IB Neuro™ (Imaging Biometrics), Olea Sphere (Olea Medical) and Mint Lesion™ (Mint Medical GmbH) software and assessed by a central reader not involved in the treatment of the patients (SB).*

- Two patients were <50 years old so how was screening for IDH mutation done – IHC or was next generation sequencing performed?

If patients were younger than 54 years, they received immunohistochemistry screening, but were routinely assessed additionally by pyrosequencing for IDH1 and IDH2. All of said analyses confirmed consistently the diagnosis of an IDH-wildtype (please see also **reviewer #1, point #1**).

A corresponding statement was added to the Methods section:

Page 10, paragraph 3: *All patients were neuropathologically confirmed as glioblastoma, IDH-wildtype (CNS WHO grade 4) according to the WHO classification for CNS tumors 2021 by immunohistochemistry (IHC). If patients were ≤ 54 years of age, they were assessed additionally by pyrosequencing for IDH1 and IDH2.*

Minor Concerns

- The sentence “Given that endothelial cells showed the highest relative CXCL12 positivity at roughly twelve times lower total cell numbers compared to the glioma cells...” (line 179-180) is confusing and hard to understand.

We rephrased the sentence accordingly:

Page 7, paragraph 4: *Both endothelial cells (E12) and glioma cells (G12) showed a significant correlation with PFS. While endothelial cells showed the highest relative CXCL12 positivity, the total number of endothelial cells was roughly twelve times lower than that of glioma cells. Therefore, we reasoned that combining E12 and G12 with approximately equal weights could embrace independent biologic mechanisms and, thus, improve the correlation with NOX-A12 treatment responses.*

- Was the SOC cohort a contemporaneous cohort? Please include the years during which the SOC cohort was identified.

The SOC cohort consisted of patients treated at our university hospital between 2010 and 2023. 20 of these patients received their first diagnosis between 2015 and 2023, while 2 were first diagnosed between 2010 and 2015. All of the patients received contemporaneous SOC treatment consisting of radiotherapy and optionally temozolomide.

We added the information about the time frame of SOC enrolment to the Methods section of the manuscript:

Page 13, paragraph 1: *To benchmark tissue and outcome, we established a reference cohort of GBM patients treated outside of the study with SOC RT and temozolomide (TMZ) at the University Hospital Bonn between 2010 and 2023.*

Reviewer #3

Overall summary

This is a well written report of the safety and outcome of the first 10 patients enrolled onto the Phase 1 dose escalation component of the GLORIA trial of a CXCL12-neutralizing aptamer given in combination with radiation. The authors note promising outcomes in this rather poor prognosis subgroup of glioblastoma and identify a subgroup responsive to this therapy.

We thank the reviewer for the positive recognition of our work and the appreciated suggestions.

Comments

1) Safety - 10 patients were enrolled onto 3 dose levels. It is unclear what dose escalation methodology was used to inform dose escalation (3+3 or Bayesian or other) but this will need clear description, particularly given safety was the primary outcome of the the Phase 1 component of the trial reported. Methodology and DLT criteria will need to be provided in a revision.

The design of the dose escalation is a traditional 3+3 rule-based design. For more details, please see also **reviewer #1, point #2**. Details on the methodology of the study design can be found in detail in the Method sections now.

We also added a definition of DLT criteria to the Methods:

Page 10, paragraph 2: *DLTs according to the Common Terminology Criteria for Adverse Events (CTCAE, version 5.0) were defined as any grade 3-4 non-hematological toxicities (excluding grade*

3 vomiting and/or nausea, if encountered without adequate and optimal prophylactic therapy), at any DL, assessed by the Investigator and/or the sponsor as related to NOX-A12.

The authors provide a list of treatment emergent adverse events in the supplemental data, however this table need clarification. I note 7 events of seizure, and 2 events of status epileptus in the data listing; alongside events of brain oedema, and psychomotor disturbance. The authors will need to clarify why these events were not considered 'possibly' related to novel treatment, given the temporal occurrence and how the authors concluded that NOX-A12 did not contribute to neurological AEs. It would be more helpful to provide tables of listings of TEAE and TRAE per dose level, and commenting specifically on AEs that are occurring >5%.

We added additional datasets for adverse events as supplementary files. For readability, we only provide summarized results in Fig. 2. Statements regarding TEAE and TRAEs were added to the Results section referring to a new Extended Data Fig. and Extended Data Tables for details.

Page 5, paragraph 3 (Results section): *The most common treatment-emergent AEs (TEAEs, n=160) were headaches which had been reported for a total of six patients with a maximum grading of grade 2 (Extended Data Fig. 1a). Increase of the alanine aminotransferase was the only treatment-related AE (TRAE) that occurred in three patients and did not exceed grade 2 (Extended Data Fig. 1b). Complete AE listings are summarized in Extended Data Table 1. TEAEs and TRAEs are provided in full in Extended Data Table 2 to Extended Data Table 5, respectively.*

Page 24, Extended Data Fig. 1: *Treatment-emergent AEs (TEAEs) and treatment-related AEs (TRAEs) in the GLORIA trial. a, Bar plots depicting percentage of affected GLORIA patients (n=10) and gradings for all TEAEs affecting at least two patients or having been reported as serious adverse event (SAE, indicated by an asterisk (*)). b, Bar plots depicting percentage of affected GLORIA patients (n=10) and gradings for all TRAEs. SAEs are indicated additionally by an asterisk (*). TRAEs are defined as TEAEs related to NOX-A12 and/or underlying disease and/or irradiation. ALT: alanine aminotransferase; GGT: gamma-glutamyltransferase.*

Page 32, Extended Data Table 2: *Complete listing of treatment-emergent adverse events per dose level.*

Page 36, Extended Data Table 3: *Complete listing of treatment-emergent adverse events related to NOX-A12 and/or underlying disease and/or irradiation per dose level.*

Page 37, Extended Data Table 4: *Complete listing of treatment-emergent adverse events related to NOX-A12 only per dose level.*

Page 38, Extended Data Table 5: *Complete listing of treatment-emergent adverse events related to underlying disease and/or irradiation per dose level.*

All AEs were reported by the individual investigator responsible for the particular clinical situation presented to him/her. In fact, there was a clear coincidence of the AEs questioned with either a simultaneous radiographic deterioration of the corresponding patients leading to an attribution of the AE to the progressing tumor, or an association with the baseline tumor volume.

In detail:

Of the six seizures reported, four occurred in patient C1-002, who had a residual tumor in the left frontal lobe of approximately 34 cm³. She already presented with seizures at baseline which deteriorated over the course of her disease as the MRI was suggestive of recurrent disease. She was hospitalized experiencing further seizures before she deceased. All of the events were considered related to the underlying disease.

Patient C1-003 had one seizure reported which also occurred at a time point of evident tumor progression in the MRI scan and prompted the investigators to conclude relation to the tumor.

Patient C3-003 suffered from a seizure while she was waiting for her follow-up MRI scan to be performed. The scan later revealed progressive disease. The event was thus also considered related to the underlying disease.

We observed a singular event of brain oedema occurring at the end of radiotherapy in patient C3-002. The oedema resolved under dexamethasone treatment and did not occur again even though the medication was discontinued while treatment with NOX-A12 continued. Therefore, the investigators deemed the event related to irradiation.

Psychomotor hyperactivity was only recorded in C1-001. At the onset of said AE, the 71-year old patient presented at the study visit with dry cough, subfebrile temperature and reduced general condition under cortisone treatment. He additionally presented with a novel psycho-motoric agitation leading to disturbance of sleep. CRP was elevated. Hospitalization was initiated to find an infection focus and to manage and clarify the psycho-motoric agitation. A CT scan revealed atypical pneumonia. The agitation was diagnosed as delir. After adequate antibiotic treatment, the patient recovered from both the infection and the neurological disorder.

2) Pharmacokinetics - the authors do not comment on brain penetration of the aptamer, at least from preclinical data. Is there expected variability from the 3 dose levels tested that may impact efficacy?

NOX-A12's mode of action is to disrupt CXCL12-dependent recruitment of circulating bone-marrow derived cells to the irradiated tumor where they contribute to restoration of the vasculature and promote the growth of surviving tumor cells. In order to achieve this, neutralization of CXCL12 in the vasculature is required, in particular of CXCL12 bound to glycosaminoglycans on the surface of endothelial cells present in tumor tissue which are the interface between blood and tumor. Tissue penetration is thus not necessary to block migration of cells between bone marrow and tumor. NOX-A12 has been shown to detach and sequester CXCL12 from cell surfaces and prevent recruitment of monocytic cells into tumor tissue following hypoxia induced by both radiotherapy and anti-VEGF therapy. As demonstrated in Fig 2a, we accordingly observed a 1,000-fold excess of CXCL12 plasma levels in our trial patients. Thus, we show here for the first time in humans the functioning mode of action of NOX-A12.

Finally, the experimentally determined production rate of CXCL12 in healthy human subjects (approx. 26 nM/h in plasma) is equivalent to the infusion rate of NOX-A12 at the lower tested dose of 200 mg/week (approx. 27 nM/h). It is thus possible that the maximal effect is already reached at this dose.

We felt these are important aspects that may be of interest to many readers and thus added the following statements to the Discussion section including also references:

Page 9, paragraph 1: *Insufficient crossing of the blood-brain-barrier is a frequent limitation of novel drugs targeting brain tumors [Wu et al.]. However, tissue penetration is not a prerequisite for NOX-A12 efficacy. NOX-A12 neutralizes CXCL12 in the blood and it also releases and sequesters CXCL12 bound to glycosaminoglycans on the surface of tumor endothelial cells at the interface between blood system and tumor cells [Hoellenriegel et al.]. Thereby, NOX-A12 disrupts CXCL12-dependent recruitment of circulating BMDCs to the hypoxic tumor tissue which prevents restoration of the tumor vasculature and hence restrains tumor cell growth in preclinical models [Chernikova et al.; Deng et al.]. Our study shows for the first time in humans that NOX-A12 treatment indeed detaches and sequesters CXCL12, as we detected a massive accumulation of the chemokine in the plasma reaching concentrations in the low micromolar range.*

3) Efficacy - the study is laudable in recruiting a poor prognosis group of patients with incompletely resected GBM, with unmethylated MGM. It is noted that 10 patients is an extremely small number to be drawing more than exploratory conclusions, however given the poor prognosis of patients, it does appear to be promising that 8/9 has some response in target lesions. Line 129 - there is no established radiological criteria to define PR in non-target lesions so this sentence appears to be overcalling and should be removed.

We agree that there is no established definition for PR of NTLs. Instead of removing, we rephrased this sentence, because we still find it noteworthy that in 7/10 patients a size reduction $\geq 50\%$ was achieved for NTLs.

Page 6, paragraph 1: *All three patients of DL 1 and all four of DL 3 reached $\geq 50\%$ size reduction of at least one non-target lesion (NTL).*

4) Biomarker translational work - the sample size is small for conclusive subgroup analyses, however the study does convincingly demonstrate proof of concept for mechanism of action with patients with higher expression of CXCL12 benefitting more.

While indeed just providing a small sample size due to the nature of the trial phase reported, we believe that our work may act as a proof of concept that justifies further investigation in larger future cohorts, both regarding efficacy of the drug but also further assessment of the EG12 score.

Overall, this manuscript warrants publication with minor corrections as described above. The authors will need to provide study protocol, along with dose finding schema and DLT criteria and expand on the safety and tolerability description as above.

We thank the reviewer for this favorable assessment of our work. As explained in detail above, all of the requested changes were performed.

The study protocol is provided alongside the publication in a minimally redacted version taking into account the interests of the sponsor:

Page 41, Suppl. Data 4: *Study protocol, minimally redacted to account for data protection rights of the sponsor.*

Which dose level is being taken forward into expansion?

According to the DSMB decision based on the data provided in this manuscript, the NOX-A12 dose of 600 mg/week was safe and well tolerated and is therefore the recommended phase II dose and also the DL being taken forward into expansion.

Page 9, paragraph 3: *The highest DL of 600 mg NOX-A12/week was safe and well tolerated. It is therefore the recommended phase II dose and also the DL being taken forward into expansion.*

The proof of concept and preliminary efficacy in a very poor prognosis subset of GBM is exciting and warrants further exploration.

Thank you again very much for this very positive evaluation of our work.

Reviewers' Comments:

Reviewer #1:

Remarks to the Author:

Thank you for responding to all my comments.

Reviewer #2:

Remarks to the Author:

The authors have addressed most of my comments. The one lingering question is related to the SOC cohort. In their response, the authors state "All of the patients received contemporaneous SOC treatment consisting of radiotherapy and optionally temozolomide."

However, the manuscript was revised to state "To benchmark tissue and outcome, we established a reference cohort of GBM patients treated outside of the study with SOC RT and temozolomide (TMZ) at the University Hospital Bonn between 2010 and 2023."

Was the temozolomide "optional" as implied in the response or did everyone receive it as implied in the revised manuscript? If only a subset actually received temozolomide, please include that number in the manuscript

Reviewer #3:

Remarks to the Author:

Many thanks for the revision and clarification provided in response to comments.

With regards the dose escalation design of 3+3, it is unclear why DL3 only included 4 patients as the model would have predicted that this should have been expanded to 6 (3+3). What was the rationale of not completing the dose escalation per. this design? Were additional patients excluded from the manuscript for reasons of evaluability - this will need to be made very clear in the CONSORT diagram with clear reasoning for the change from the dose escalation model particularly given safety is the primary objective of this Phase 1 trial.

We would like to thank the reviewers and the editors for taking the time to thoroughly review our manuscript and for their positive response to our rebuttal letter.

We addressed all the remaining comments and suggestions in our point-by-point response below. All additional changes to the manuscript are color highlighted. We are happy to address any further questions that may arise.

Reviewer #2

The authors have addressed most of my comments. The one lingering question is related to the SOC cohort. In their response, the authors state "All of the patients received contemporaneous SOC treatment consisting of radiotherapy and optionally temozolomide." However, the manuscript was revised to state "To benchmark tissue and outcome, we established a reference cohort of GBM patients treated outside of the study with SOC RT and temozolomide (TMZ) at the University Hospital Bonn between 2010 and 2023." Was the temozolomide "optional" as implied in the response or did everyone receive it as implied in the revised manuscript? If only a subset actually received temozolomide, please include that number in the manuscript

We thank the reviewer for this positive feedback. As we only considered patients lacking MGMT-methylation eligible for the SOC cohort, temozolomide was optional. Thus, 4/22 patients of the SOC cohort did not receive temozolomide. This information is additionally provided in Extended Data Table 6. To clarify, we made the following modification to the Methods section of the manuscript:

Page 13, paragraph 1: *To benchmark tissue and outcome, we established a reference cohort of GBM patients treated outside of the study with SOC RT and optional temozolomide (TMZ) at the University Hospital Bonn between 2010 and 2023.*

Page 13, paragraph 1: *Clinical criteria were: ECOG of 0-2, status post biopsy or incomplete resection and first line therapy with RT (and optionally TMZ, n=18/22 receiving TMZ).*

Many thanks for the revision and clarification provided in response to comments. With regards the dose escalation design of 3+3, it is unclear why DL3 only included 4 patients as the model would have predicted that this should have been expanded to 6 (3+3). What was the rationale of not completing the dose escalation per. this design?

Per design, all DLs contained three patients, unless DLTs required addition of another three patients at the next higher DL. The reason for the inclusion of a fourth patient in DL3 was not concerns regarding safety, but a patient that had withdrawn consent and dropped out of the trial. Since this patient was part of the highest dose level cohort and had only received treatment for a total of roughly three months and also required treatment interruptions due to an unrelated sigma diverticulitis, the sponsor decided to replace this patient to ensure safety data quality. Thus, the final ITT cohort of this DL consisted of 4 patients. We have indicated this in the CONSORT diagram in Fig. 1b ("premature treatment discontinuation"). To make the proceedings clearer, we have extended the descriptions in the manuscript and in the patient narrative (Suppl. Data 3):

Page 5, paragraph 2 (Results): *Between September 2019 and September 2021, three patients were enrolled at each DL. One patient of DL 3 dropped out early and was replaced to ensure safety data quality, hence a total of ten patients were treated with RT and NOX-A12 (Fig. 1b).*

Suppl. Data 3 (Patient narratives, page 3, C3-002): *On June 24, 2021 another unscheduled scan was suggestive for tumor recurrence (pPD) and NOX-A12 was discontinued. The patient then refused any further scan and intervention and died on August 14, 2021. The patient had as part of the highest dose level cohort only received treatment for a short period and with various treatment interruptions. While the patient remained part of the intention-to-treat cohort, he was therefore considered a drop-out and replaced to ensure data quality of the primary endpoint (safety).*

Were additional patients excluded from the manuscript for reasons of evaluability - this will need to be made very clear in the CONSORT diagram with clear reasoning for the change from the dose escalation model particularly given safety is the primary objective of this Phase 1 trial.

There were no additional patients excluded. All DL patients that received the treatment are being reported in this manuscript. The decision to include another patient in DL3 was particularly driven by the desire to ensure safety data quality in the high dose-cohort of this trial.

Reviewers' Comments:

Reviewer #2:

Remarks to the Author:

The authors have addressed all my comments.

Reviewer #3:

Remarks to the Author:

Many thanks for your response to our comments.

The traditional 3+3 design defines the MTD as the highest dose level in which six patients were treated and, at most, one patient experienced a DLT during the first cycle of therapy. (Le Tourneau C, Lee JJ, Siu LL. Dose escalation methods in phase I cancer clinical trials. *J Natl Cancer Inst.* 2009;101:708-720; Ivy SP, Siu LL, Garrett-Mayer E, et al. Approaches to phase 1 clinical trial design focused on safety, efficiency, and selected patient populations: a report from the clinical trial design task force of the national cancer institute investigational drug steering committee. *Clin Cancer Res.* 2010;16:1726-1736.)

It appears from the authors response that they halted the expansion of the DL3 cohort after just three patients were recruited given no DLT was seen in this cohort. This is unusual, but if pre-specified in the protocol will just need to be clarified as a modified 3+3 design, but in the event no DLT was seen at the highest level, this would be declared the recommended dose after 3 patients without expansion to 6.

We would like to thank the reviewers and the editors for taking the time to thoroughly review our manuscript and for their positive response to our rebuttal letter.

We addressed the remaining comment following the reviewer's suggestion in our point-by-point response below. All additional changes to the manuscript are color highlighted. We are happy to address any further questions that may arise.

Reviewer #3

Many thanks for your response to our comments.

The traditional 3+3 design defines the MTD as the highest dose level in which six patients were treated and, at most, one patient experienced a DLT during the first cycle of therapy. (Le Tourneau C, Lee JJ, Siu LL. Dose escalation methods in phase I cancer clinical trials. *J Natl Cancer Inst.* 2009;101:708-720; Ivy SP, Siu LL, Garrett-Mayer E, et al. Approaches to phase 1 clinical trial design focused on safety, efficiency, and selected patient populations: a report from the clinical trial design task force of the national cancer institute investigational drug steering committee. *Clin Cancer Res.* 2010;16:1726-1736.)

It appears from the authors response that they halted the expansion of the DL3 cohort after just three patients were recruited given no DLT was seen in this cohort. This is unusual, but if pre-specified in the protocol will just need to be clarified as a modified 3+3 design, but in the event no DLT was seen at the highest level, this would be declared the recommended dose after 3 patients without expansion to 6.

We followed the suggestion of reviewer #3 and specified the study design in the manuscript as a modified 3+3 design:

Page 5, paragraph 1: *In the dose-escalation part of the trial reported here, NOX-A12 was administered in a **modified** 3+3 rule-based design with escalating dose levels (DLs) of 200, 400 and 600 mg NOX-A12 per week*

Page 10, paragraph 2: *The dose escalation was designed as a **modified** 3+3 rule-based design according to Le Tourneau et al.⁵¹ with three successional cohorts consisting of three patients each.*

Page 16, paragraph 3: *The dose escalation was designed as a **modified** 3+3 rule-based design as described above and in the study protocol provided in Supplementary Note 2.*